# Foresee-to-Ground: From Predictive Temporal Perception to Evidence-Driven Reasoning for Video Temporal Grounding

**Zelin Zheng**[1 2]  **Xinyan Liu**[3 4]  **Ruixin Li**[1 2]  **Antoni B. Chan**[4]  **Guorong Li**[1]  **Qingming Huang**[1]  **Laiyun Qing**[1 2]

## Abstract

Current Video-LLM approaches for Video Temporal Grounding (VTG) typically rely on direct timestamp generation from an unstructured visual-token stream, often leading to brittle numerics and inconsistent boundaries. To address this, we propose Foresee-to-Ground (F2G), a framework that reformulates VTG as a verifiable Identify-then-Measure problem. F2G integrates Predictive Temporal Perception with Evidence-Driven Reasoning: it learns boundary-sensitive temporal representations to build a video-wide evidence pool of candidate event segments, and exposes these segments to the LLM as citable evidence units that bind boundary prediction to explicit event hypotheses. By decoupling event identification from precise boundary measurement, F2G stabilizes grounding and makes predictions verifiable. Extensive experiments demonstrate that F2G consistently improves grounding accuracy across diverse benchmarks, transfers robustly across different Video-LLM backbones, and preserves general video understanding capabilities. Our project is available at https://github.com/zelion2003/Foresee-to-Ground.

## 1 Introduction

Video Temporal Grounding (VTG) serves as a pivotal task in long-form video understanding, mapping open-vocabulary queries to specific temporal segments (Zhao et al., 2017; Wang et al., 2019; Li et al., 2020; Nan et al., 2021; Zhang et al., 2023; Wu et al., 2026). Video-LLMs have emerged

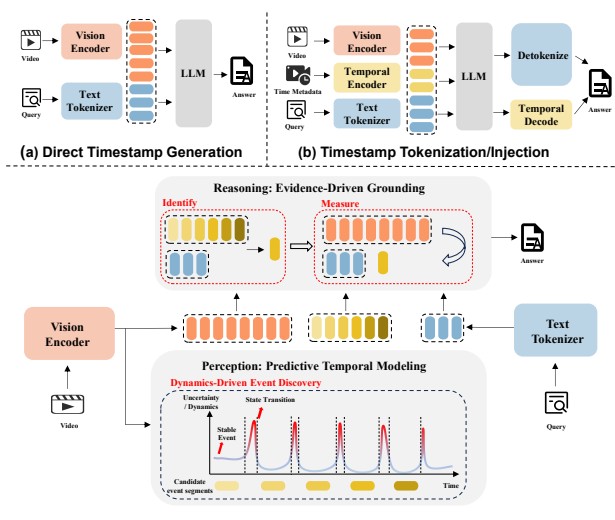

*Figure 1.* **Paradigms for VTG with Video-LLMs.** (a) Direct timestamp generation; (b) Direct timestamp generation with time-text interface; (c) F2G: a verifiable grounding pipeline.

as the predominant backbone for VTG. However, the prevailing paradigm treats grounding as a direct timestamp regression problem. This formulation is architecturally misaligned, as it compels LLMs to map flattened visual tokens onto continuous coordinates from within a discrete representation space. Such a black-box regression approach inevitably leads to brittle inference and high-variance temporal boundaries.

These regression-based approaches are conceptually incongruent: they compel LLMs to map a discrete token space onto a continuous temporal domain. While existing methods mitigate this via timestamp discretization (Huang et al., 2024a; Qian et al., 2024; Ren et al., 2024; Guo et al., 2025b), they largely overlook the underlying cognitive process of temporal grounding. Humans do not ground temporal queries by treating videos as a continuous stream of metric time; instead, when grounding a query, we first make an explicit segment commitment (identifying the event), then refine its precise boundaries (measuring the interval). This motivates an Identify-then-Measure formulation, reframing VTG from black-box numeric regression into a structured grounding process where explicit evidence selection guides evidence-conditioned boundary refinement. Fig. 1 summa-

---

[1]School of Computer Science and Technology, University of Chinese Academy of Sciences, Beijing, China [2]Beijing Key Laboratory of Embodied Intelligence Computing, Beijing, China [3]Faculty of Computing, Harbin Institute of Technology, Weihai, China [4]City University of Hong Kong, Hong Kong, China. Correspondence to: Laiyun Qing <lyqing@ucas.ac.cn>.

*Proceedings of the 43rd International Conference on Machine Learning*, Seoul, South Korea. PMLR 306, 2026. Copyright 2026 by the author(s).

rizes these paradigms.

To operationalize this intuition, we propose Foresee-to-Ground (F2G), a framework that translates the Identify-then-Measure routine into a verifiable grounding pipeline. F2G bridges temporal perception and LLM reasoning by constructing an explicit, video-wide evidence pool of candidate event segments. Each evidence unit contains a span identifier, a coarse temporal interval, and segment-local visual evidence, and is presented to the LLM as a discrete, citable event hypothesis. The LLM then identifies the most relevant hypothesis and measures precise temporal boundaries through evidence-conditioned refinement.

F2G builds this evidence pool with Predictive Temporal Perception. Motivated by the human tendency to perceive videos as coherent events rather than a flat visual stream, this module decomposes untrimmed videos into a compact set of candidate segments. These candidates are extracted from boundary-aware representations learned via predictive objectives, which encourage the temporal module to capture coherent event dynamics and transition cues. The resulting evidence pool supports subsequent Evidence-Driven Reasoning, where the LLM input is augmented with indexed evidence units. By binding each boundary prediction to a specific evidence ID, F2G shifts grounding from unconstrained timestamp regression over a flattened visual-token stream to verifiable, citation-based inference.

We instantiate F2G on Qwen3-VL-8B-Instruct (Bai et al., 2025a) and evaluate it on standard VTG benchmarks. Across settings, F2G improves grounding accuracy while markedly reducing the instability under repeated inference. Moreover, the same *+F2G-FT* recipe transfers to other Video-LLMs with minimal changes, consistently boosting VTG without sacrificing general video understanding. More broadly, F2G suggests a general recipe for applying LLMs to structured perception: make discrete commitments explicit, and reserve continuous prediction for local refinement.

More broadly, F2G highlights a simple design principle for Video-LLM grounding: expose intermediate evidence explicitly before asking the model to produce precise temporal boundaries. Our contributions are summarized as follows:

1. We introduce Foresee-to-Ground (F2G), a VTG framework that makes segment-level evidence explicit and enables verifiable grounding through supervised evidence citation.
2. We develop Predictive Temporal Perception, which learns boundary-sensitive temporal representations and extracts a compact, ranked evidence pool of candidate event segments from untrimmed videos.
3. We propose Evidence-Driven Reasoning, which augments the LLM input with citable evidence units and implements Identify-then-Measure through joint span-ID and timestamp supervision.

## 2 Related Work

### 2.1 Video Temporal Grounding with Video-LLMs

Video Temporal Grounding (VTG) grounds the temporal interval of a language query in an untrimmed video, typically by predicting start/end timestamps (Zhao et al., 2017; Wang et al., 2019; Li et al., 2020; Nan et al., 2021; Li et al., 2025d), covering tasks such as moment retrieval (Gao & Xu, 2021; Xu et al., 2022; Foo et al., 2023; Zala et al., 2023; Li et al., 2025a; Wang et al., 2025a), dense video captioning (Krishna et al., 2017; Yang et al., 2023; Cao et al., 2024; Kim et al., 2024), and highlight detection (Lei et al., 2021; Ren et al., 2024). Motivated by open-vocabulary generalization and a unified instruction-following interface (Li et al., 2025b; Liu et al., 2023; Wang et al., 2024a; Yi et al., 2024; Zhang et al., 2025b), recent works increasingly build VTG on top of Video-LLMs to solve multiple grounded tasks within a single generative framework. A persistent challenge is that Video-LLMs must map a long, flattened visual-token stream to metric time, making timestamp decoding numerically brittle and unstable. To bridge this gap, prior adaptations largely fall into two complementary lines.

**Timestamp tokenization and injection:** discretizing time into bins/tokens (Qian et al., 2024; Guo et al., 2025b) or injecting explicit temporal cues such as timestamps (Huang et al., 2024a; Ren et al., 2024; Zeng et al., 2025) and visual indices/number prompts (Wu et al., 2025). These methods improve the time-to-text interface, but the model still performs a direct stream-to-time mapping. **Segment-level structural bias:** including event-structured decoding (Guo et al., 2025b), segment-level fine-tuning (Hu et al., 2025), and coarse-to-fine multi-pass prompting (Nie et al., 2024). While such structure can help localization, it often increases inference cost, and the intermediate segment hypothesis is not always explicit, supervised, or directly attributable. In contrast, Foresee-to-Ground introduces an evidence-mediated interface that turns segment hypotheses into a compact, video-wide evidence pool of candidate event segments. Each evidence unit is associated with a citable span ID, a coarse temporal interval, and segment-local visual evidence, so the intermediate hypothesis is not merely a narrowed search range but a supervised and attributable event-level variable. The model then grounds the query by identifying one such evidence unit and measuring precise metric boundaries via evidence-driven reasoning under the cited hypothesis.

### 2.2 Self-Supervised Video Representation Learning

Self-supervised video representation learning acquires transferable spatiotemporal features from unlabeled videos via pretext objectives (Xu et al., 2019; Benaim et al., 2020; Huang et al., 2021; Qian et al., 2021). Masked reconstruction methods recover missing spatiotemporal content and

scale well to large corpora (Tong et al., 2022; Wang et al., 2023), while recent predictive approaches replace pixel-level synthesis with latent-space prediction to capture temporal dynamics more compactly (Bardes et al., 2024; Assran et al., 2025). At the same time, predictive training can be sensitive to optimization and latent geometry, motivating regularization that stabilizes embedding distributions and improves robustness (Balestriero & LeCun, 2025). Unlike prior work that treats such pretraining mainly as a transferable representation, we use it as predictive temporal perception for verifiable VTG: the learned features are optimized to discover event segments and construct an explicit evidence pool for Identify-then-Measure grounding.

## 3 Foresee-to-Ground

We formulate VTG as a verifiable structured prediction problem under an explicit Identify-then-Measure formulation:

$$
\begin{aligned}
p(A, T, z \mid V, Q, \mathcal{S}_K(V)) = {} & p(z \mid V, Q, \mathcal{S}_K(V)) \\
& \cdot p(A, T \mid z, V, Q, \mathcal{S}_K(V)).
\end{aligned}
\tag{1}
$$

where $V$ and $Q$ denote the input video and query, $T = (t^{\mathrm{st}}, t^{\mathrm{ed}})$ is the metric interval, and $A$ is an grounded answer. Crucially, $z \in \{1, \ldots, K\}$ indexes a single candidate segment from a compact, video-wide evidence pool $\mathcal{S}_K(V)$.

The first factor predicts the cited index $z$ (Identify), and the second factor generates $(A, T)$ conditioned on the cited hypothesis (Measure), turning grounding into a verifiable decision rather than direct timestamp regression. This factorization is implemented as a single structured response rather than two separate LLM passes. All evidence units are provided in the same context, while the output is supervised to contain exactly one evidence citation that makes the generated interval attributable to a specific hypothesis.

As illustrated in Fig. 2, F2G realizes this framework through a synergistic design of predictive temporal perception and evidence-driven reasoning. The perception module extracts boundary-sensitive temporal features $U$, proposes Top-$K$ candidate event segments, and aggregates segment-local visual evidence via a Span Evidence Encoder (SEE), forming the video-wide evidence pool $\mathcal{S}_K(V)$. Conditioned on the instruction, video tokens, and $\mathcal{S}_K(V)$, the Video-LLM first cites one evidence ID $z$ and then generates the final boundaries $T$ and answer $A$ under the cited hypothesis.

We optimize F2G via a three-stage curriculum: i) pretraining predictive temporal perception, ii) warming up proposal generation, and iii) fine-tuning the Video-LLM for evidence-driven grounding.

### 3.1 Predictive Temporal Perception: Dynamics-Driven Event Discovery

In Stage-1, we pretrain a temporal module to produce boundary-sensitive representations that are well suited for event discovery in untrimmed videos. Rather than regressing temporal boundaries directly, this stage learns dynamics-aware features from which coherent event hypotheses can be efficiently extracted. We achieve this via a multi-view latent prediction objective (Bardes et al., 2024; Assran et al., 2025; Balestriero & LeCun, 2025).

Given an untrimmed video $V$ sampled into $N$ time steps, the Video-LLM's vision encoder produces spatially dense tokens and a projector maps them into the Video-LLM embedding space:

$$
H^v = \mathrm{Proj}(\mathrm{VisionEnc}(V)) \in \mathbb{R}^{N \times S \times D}, \tag{2}
$$

where $S$ is the number of spatial tokens per time step and $D$ matches the LLM embedding dimension. We pool over spatial tokens to form the temporal sequence shared by all stages:

$$
X = \mathrm{Pool}(H^v) \in \mathbb{R}^{N \times D}. \tag{3}
$$

Stage-1 pretrains the temporal module on multi-view samples derived from $X$, rather than directly supervising temporal boundaries. The key intuition is that within a coherent event, long-range temporal dynamics are relatively predictable from partial observations, whereas near event boundaries, the same partial evidence can correspond to multiple plausible continuations. By predicting global temporal dynamics from local views, the model is encouraged to distinguish stable within-event evolution from boundary-induced ambiguity:

$$
\min_g \ \mathbb{E}\big[\|U_g - g(U_l)\|_2^2\big], \tag{4}
$$

where $U_g$ and $U_l$ denote global-view and local-view latents in a simplified two-view case, and $g$ denotes the predictor.

Concretely, we construct multiple views on the same temporal sequence $X \in \mathbb{R}^{N \times D}$. The global view $X_g \in \mathbb{R}^{N_g \times D}$ preserves full-range temporal context, while each local view $X_l^{(v)} \in \mathbb{R}^{N_v \times D}$ contains partial temporal evidence obtained by a view-specific crop, stride, or subsampling pattern. Here, $N_g$ and $N_v$ denote the numbers of timesteps in the global and local views, respectively. All views share the same temporal module:

$$
U_g = f_\theta(X_g), \qquad U_l^{(v)} = f_\theta(X_l^{(v)}), \ \ v \in \mathcal{V}, \tag{5}
$$

A lightweight predictor $g_\phi$ maps each local-view latent to the global target latent. We additionally use a learnable view embedding $e_v$ to indicate the local-view type (e.g., scale/stride/crop pattern):

$$
\hat{U}_g^{(v)} = g_\phi\Big(U_l^{(v)} + e_v\Big), \tag{6}
$$

and minimize a latent-space predictive loss:

$$
\mathcal{L}_{\mathrm{pred}} = \mathbb{E}\left[\sum_{v \in \mathcal{V}} \Big\|\mathrm{sg}(U_g) - \hat{U}_g^{(v)}\Big\|_2^2\right], \tag{7}
$$

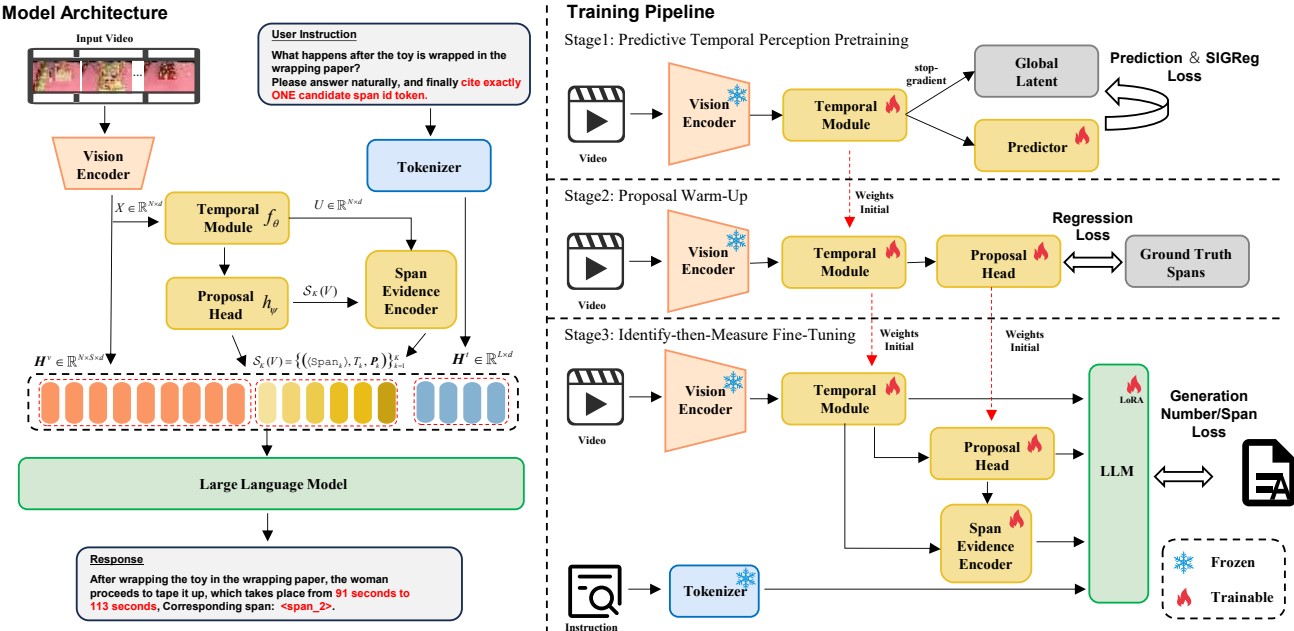

*Figure 2.* **Foresee-to-Ground framework overview.** Left: model architecture and LLM input sequence construction with evidence units. Right: three-stage training pipeline (Stage-1: predictive temporal perception pretraining; Stage-2: proposal warm-up; Stage-3 evidence-driven Identify-then-Measure fine-tuning via LoRA on the LLM).

where $\mathrm{sg}(\cdot)$ denotes stop-gradient. Since each local view $X_l^{(v)}$ contains only partial temporal context while the global view $X_g$ summarizes long-range dynamics, minimizing $\mathcal{L}_{\mathrm{pred}}$ is discouraged from collapsing to a trivial copy-and-match solution. Instead, the shared backbone is driven to encode the temporal cues that make global evolution predictable from partial evidence, which tends to highlight transition/boundary-related signals in the learned temporal features.

To stabilize the latent geometry and facilitate reliable downstream boundary refinement, we regularize the temporal representations using a sliced isotropic Gaussian regularizer (SIGReg) (Balestriero & LeCun, 2025).

Let $\tilde{u}$ denote pooled latent samples collected from $\{\mathrm{sg}(U_g)\} \cup \{U_l^{(v)}\}_{v \in \mathcal{V}}$ over a minibatch (pooling over time and batch). Let $\mathcal{A}$ be a set of random unit directions (sampled on the sphere), $\hat{p}(a^\top \tilde{u})$ the empirical 1D distribution of projected samples, and $\mathcal{N}(0, 1)$ the standard normal. We define

$$\mathcal{L}_{\mathrm{SIG}} = \mathbb{E}_{a \sim \mathcal{A}}\Big[D\big(\hat{p}\big(a^\top \tilde{u}\big), \mathcal{N}(0, 1)\big)\Big], \qquad (8)$$

where $D(\cdot, \cdot)$ is a divergence between 1D distributions. This regularizer encourages a well-conditioned latent geometry and improves optimization stability.

The overall Stage-1 objective is

$$\mathcal{L}_{\mathrm{S1}} = (1 - \lambda)\mathcal{L}_{\mathrm{pred}} + \lambda\mathcal{L}_{\mathrm{SIG}}. \qquad (9)$$

We summarize the complete optimization procedure in Alg. 1.

After Stage-1 pretraining, we discard the predictor $g_\phi$ and apply the pretrained temporal module to the full temporal sequence, obtaining $U = f_\theta(X) \in \mathbb{R}^{N \times D}$ for Stage-2 proposal generation and Stage-3 evidence construction.

### 3.2 Proposal Warm-up and Evidence Pool Construction

Stage-2 trains a lightweight proposal head $h_\psi$ to enumerate a compact, ranked set of candidate event segments from the boundary-sensitive temporal features $U = f_\theta(X)$. This warm-up stage aligns proposal generation with the downstream evidence-driven reasoning interface, ensuring that the Top-$K$ candidates form a high-recall evidence pool suitable for citation.

We implement $h_\psi$ as a lightweight self-attention stack followed by MLP regression and scoring:

$$\begin{aligned} \bar{U} &= \mathrm{MHSAStack}_\psi(U) \in \mathbb{R}^{N \times D}, \\ (\tilde{s}_i, \tilde{c}_i) &= \mathrm{MLP}_\psi(\bar{U}_i), \end{aligned} \qquad (10)$$

where $\tilde{s}_i = (\tilde{\tau}_i^{\mathrm{st}}, \tilde{\tau}_i^{\mathrm{ed}}) \in [0, 1]^2$ is a normalized segment and $\tilde{c}_i \in \mathbb{R}$ is its objectness score. Collecting all predictions gives $\tilde{\mathcal{S}}_N(V) = \{(\tilde{s}_i, \tilde{c}_i)\}_{i=1}^N$, and we keep the Top-$K$ candidates by objectness score:

$$\tilde{\mathcal{S}}_K(V) = \mathrm{TopK}\Big(\tilde{\mathcal{S}}_N(V)\Big). \qquad (11)$$

Stage-2 warm-up supervises $h_\psi$ with a small set of labeled

**Algorithm 1** Predictive Temporal Perception via Multi-view Latent Prediction

---

**Input:** temporal feature sequence $X$;
**Parameters:** shared backbone $f_\theta$; predictor $g_\phi$; view embeddings $\{e_v\}$; weight $\lambda$;
**Output:** pretrained backbone $f_\theta$ ;
**repeat**
    Sample global view $X_g$ and local views $\{X_l^{(v)}\}_{v\in\mathcal{V}}$
    Encode views: $U_g \leftarrow f_\theta(X_g); U_l^{(v)} \leftarrow f_\theta(X_l^{(v)})$
    Stop-gradient target: $\bar{U}_g \leftarrow \text{sg}(U_g)$
    Predict global latent from each local latent:
    **for** each $v \in \mathcal{V}$ **do**
        $\hat{U}_g^{(v)} \leftarrow g_\phi\left(U_l^{(v)} + e_v\right)$
    **end for**
    $\mathcal{L}_{\text{pred}} \leftarrow \sum_{v\in\mathcal{V}} \|\bar{U}_g - \hat{U}_g^{(v)}\|_2^2$
    Pool latent samples $\tilde{u} \leftarrow \text{Pool}(\{\bar{U}_g\} \cup \{U_l^{(v)}\}_{v\in\mathcal{V}})$
    $\mathcal{L}_{\text{SIG}} \leftarrow \text{SIGReg}(\tilde{u})$
    $\mathcal{L} \leftarrow (1-\lambda)\mathcal{L}_{\text{pred}} + \lambda\mathcal{L}_{\text{SIG}}$
    Update $\theta, \phi$, and $\{e_v\}$ by SGD on $\mathcal{L}$
**until** convergence

---

VTG intervals to align proposal objectness and ranking with the evidence interface. Given the ground-truth normalized interval $\tilde{s}^\star$, we train proposal head with a standard regression-and-scoring objective:

$$\mathcal{L}_{\text{S2}} = \mathcal{L}_{\text{reg}}(\{\tilde{s}_i\}, \tilde{s}^\star) + \eta\,\mathcal{L}_{\text{score}}, \qquad (12)$$

where $\mathcal{L}_{\text{score}}$ uses IoU-derived targets so high-IoU segments receive higher objectness scores. Importantly, this supervision remains query-agnostic: although Stage-2 uses VTG-style interval annotations, the proposal head does not access the language query.

After Top-$K$ selection, we map each normalized segment to metric time $T_k$ and attach segment-local visual evidence via a Span Evidence Encoder (SEE). Given a candidate interval $T_k$, we first crop the temporal features to obtain a variable-length segment sequence

$$U_k = \text{Crop}(U, T_k) \in \mathbb{R}^{N_k \times D}. \qquad (13)$$

SEE is a Q-Former style evidence encoder with $M$ learnable query tokens. It aggregates $U_k$ into a fixed-length evidence token set via stacked cross-attention:

$$P_k = \text{SEE}(U_k) = \text{MHCAStack}(B, U_k) \in \mathbb{R}^{M \times D}, \qquad (14)$$

where $B \in \mathbb{R}^{M \times D}$ denotes the learnable queries.

Finally, we construct the evidence pool

$$\mathcal{S}_K(V) = \left\{ (\langle\text{Span}_k\rangle,\ T_k,\ P_k) \right\}_{k=1}^{K}, \qquad (15)$$

where $\langle\text{Span}_k\rangle$ is a discrete citable ID, $T_k$ is the coarse metric hypothesis, and $P_k$ provides the segment evidence in

embedding space. We provide the string-level instruction, an example of the top-K proposals and example response formats in Appendix B.

### 3.3 Identify-then-Measure: Evidence-Driven Reasoning

Given the video-wide evidence pool $\mathcal{S}_K(V)$, F2G performs verifiable grounding by enforcing an explicit Identify-then-Measure routine within Video-LLM decoding. Concretely, the decoder is supervised to make an explicit commitment to one evidence unit and to generate the final metric interval $T$ and answer $A$ under this cited hypothesis within a single structured response.

With the evidence-augmented input sequence, F2G constrains decoding to cite exactly one identifier token. Let $z \in \{1, \ldots, K\}$ denote the cited index; the identification factor in Eq. 1 is implemented as next-token probabilities over the ID vocabulary:

$$p(z \mid V, Q, \mathcal{S}_K(V)) = p(\langle\text{Span}_z\rangle \mid V, Q, \mathcal{S}_K(V)). \qquad (16)$$

The cited index $z$ explicitly selects the evidence unit $(\langle\text{Span}_z\rangle, T_z, P_z) \in \mathcal{S}_K(V)$ and serves as the hypothesis passed to boundary generation.

Conditioned on the cited evidence $(T_z, P_z)$, the model then outputs the final metric interval $T$ and $A$ within the same structured response. By tying timestamp generation to an explicit evidence citation, boundary prediction is associated with an event-segment hypothesis rather than performed as unconstrained timestamp regression from a flattened token stream; this reduces decoding ambiguity and improves stability across repeated generations.

In Stage-3, we adapt the Video-LLM reasoning stack by fine-tuning the LLM via LoRA (Hu et al., 2022) and simultaneously train the span evidence encoder (SEE) that produces $\{P_k\}$. Given a ground-truth interval $T^\star$, we assign the citation target $z^\star$ as the best-overlapping candidate in the pool,

$$z^\star = \arg\max_{k\in\{1,\ldots,K\}} \text{IoU}(T_k, T^\star), \qquad (17)$$

and optimize the evidence-driven objective:

$$\mathcal{L}_{\text{S3}} = \mathcal{L}_{\text{LM}} + \alpha\,\mathcal{L}_{\text{id}} + \beta\,\mathcal{L}_{\text{time}}, \qquad (18)$$

where $\mathcal{L}_{\text{LM}}$ is the sequence cross-entropy over the structured output, $\mathcal{L}_{\text{id}}$ supervises the cited identifier token $\langle\text{Span}_{z^\star}\rangle$, and $\mathcal{L}_{\text{time}}$ supervises the emitted numeric timestamp tokens for $(t^{\text{st}}, t^{\text{ed}})$. Because Identify-then-Measure conditions decoding on $\mathcal{S}_K(V)$, evidence quality upper-bounds attainable grounding accuracy; we therefore keep the temporal module and proposal head $(f_\theta, h_\psi)$ trainable with a smaller learning rate and add a lightweight proposal loss to maintain alignment:

$$\mathcal{L}_{\text{S3+}} = \mathcal{L}_{\text{S3}} + \gamma\,\mathcal{L}_{\text{reg}}, \qquad (19)$$

where $\mathcal{L}_{\text{reg}}$ reuses the Stage-2 proposal objective's (Eq. 12) regression term, and $\gamma$ is set small so optimization remains centered on evidence-conditioned decoding while the evidence pool continues to improve.

# 4 Experiments

## 4.1 Experiment Setup

We evaluate F2G on VTG benchmarks spanning moment retrieval (Charades-STA (Gao et al., 2017), ActivityNet Captions (Krishna et al., 2017)) and highlight detection (QVHighlights (Lei et al., 2021)). We employ a 220K-sample VTG instruction fine-tuning dataset, following the setup in prior grounding studies (Huang et al., 2024a; Wu et al., 2025). It is built from 70K QA pairs in DiDeMo (Mithun et al., 2019) and ActivityNet Captions (Krishna et al., 2017) plus additional grounding instructions from VTimeLLM (Huang et al., 2024a). The training set includes ActivityNet Captions videos but excludes Charades-STA and QVHighlights, so results on the latter are reported as zero-shot transfer. For clarity, we summarize the stage-wise training data and implementation details in Appendix C.

We use Qwen3-VL-8B-Instruct (Bai et al., 2025a) as the base model and follow the three-stage pipeline in Fig. 2. Stage-3 applies LoRA (Hu et al., 2022) to the LLM for 1 epoch (global batch 64) with AdamW (Kingma & Ba, 2015) and cosine decay (peak LR $10^{-4}$, warm-up 0.05; $r{=}64$, $\alpha{=}128$). We sample videos at 1 FPS and use Top-$K{=}8$ candidates with 4 SEE queries per sample; other settings follow the default Qwen3-VL configuration. All experiments run on 4×A6000 GPUs, and the instruction fine-tuning dataset details are in Appendix D.

For moment retrieval, we report mIoU and Recall@1 at IoU threshold $m$ (R@$m$), $m \in \{0.3, 0.5, 0.7\}$, with IoU computed between predicted and ground-truth intervals. For highlight detection, we report mAP and HIT@1, where HIT@1 tests whether the top-ranked clip overlaps a ground-truth highlight segment. All metrics follow standard VTG practice (Huang et al., 2024a; Qian et al., 2024; Ren et al., 2024; Wang et al., 2024b; Guo et al., 2025a).

## 4.2 Main Results

Prior Video-LLM VTG methods are typically built on different backbones. Since foundation models evolve rapidly, their reported results are not strictly backbone-controlled. Accordingly, we do not claim strict comparability across backbones; instead, we report each method with its backbone for transparency, and provide a backbone-controlled and dataset-controlled comparison on Qwen3-VL-8B to quantify method gains.

**Comparison with Prior VTG Methods.** Table 1 shows that *+F2G-FT* achieves strong results among reported

Video-LLM VTG methods, while the backbone-controlled comparison on Qwen3-VL-8B confirms clear gains over conventional fine-tuning. The gains concentrate at stricter IoU thresholds, indicating improved boundary sharpness from evidence-conditioned Identify-then-Measure decoding rather than merely higher coarse recall. Notably, the same recipe improves both zero-shot transfer and in-domain evaluation, suggesting that explicit event segment evidence and local refinement generalize across datasets. On QVHighlights, *+F2G-FT* improves both mAP and HIT@1 (Table 1), showing that the evidence-mediated interface benefits query-conditioned relevance ranking in addition to temporal localization. Together, these results support F2G as a unified recipe across heterogeneous VTG formulations.

**Effectiveness across Video-LLMs.** Table 2 shows that F2G transfers robustly across Video-LLM backbones: the same *+F2G-FT* recipe consistently improves VTG when applied to widely used open-source models, including LLaVA-NeXT-7B (Li et al., 2025c), Qwen2.5-VL-7B (Bai et al., 2025b), and Qwen3-VL-8B (Bai et al., 2025a), on both Charades-STA and ActivityNet-Captions. Overall, these results position F2G as a generally applicable VTG adaptation recipe rather than a backbone-specific modification.

**Generalization to TimeLens-Bench.** To further evaluate F2G under a stricter fine-grained VTG protocol, we conduct experiments on TimeLens-Bench (Zhang et al., 2025a). Following the TimeLens setting, we keep Stage-1/2 unchanged and replace only the Stage-3 instruction data with TimeLens-100K (Zhang et al., 2025a). Table 3 reports the results on Charades-TimeLens, ActivityNet-TimeLens, and QVHighlights-TimeLens. F2G consistently improves over the Qwen3-VL-8B baseline and remains competitive with TimeLens-specialized models, while using pure SFT without RL-based post-training. These results suggest that the evidence-mediated Identify-then-Measure formulation transfers beyond conventional VTG benchmarks.

**Qualitative Results.** Fig. 3 shows two examples illustrating F2G's Identify-then-Measure behavior and compares it with Qwen3-VL (+FT) and two prior Video-LLM VTG methods (TimeChat and VTimeLLM). We visualize the ground truth interval, the proposal interval cited by F2G, and the final interval measured under the cited evidence, along with predictions from the compared baselines. In Case A (stable visual content), TimeChat and VTimeLLM drift to partial sub-events, while Qwen3-VL (+FT) truncates the long action. In contrast, F2G first cites an evidence interval that covers the correct event region and then measures boundaries closer to the ground truth. In Case B (rapid scene changes), distractor transitions cause the baselines to over-extend across irrelevant segments. F2G remains confined to the cited evidence neighborhood and measures a tighter interval, improving temporal precision.

*Table 1.* **Comparison on VTG benchmarks with previous state-of-the-art methods.** † denotes zero-shot transfer evaluation. *+FT* and *+F2G-FT* denote conventional fine-tuning and Foresee-to-Ground fine-tuning on the same instruction set, respectively. The best and second-best results in each column are highlighted in **bold** and underline, respectively.

| Method | Backbone | Charades-STA† | | | | ActivityNet-Captions | | | | QVHighlights† | |
|---|---|---|---|---|---|---|---|---|---|---|---|
| | | R@0.3 | R@0.5 | R@0.7 | mIoU | R@0.3 | R@0.5 | R@0.7 | mIoU | mAP | HIT@1 |
| GroundingGPT (Li et al., 2024) | CLIP ViT-L/14 + Vicuna-v1.5-7B | - | 29.6 | 11.9 | - | - | - | - | - | - | - |
| LITA (Huang et al., 2024b) | CLIP ViT-L/14 + Vicuna-7B | - | - | - | - | - | 25.9 | - | 28.6 | - | - |
| VTG-LLM (Guo et al., 2025a) | ViT-G/14 + BLIP + LLaMA-2-7B | 52.0 | 33.8 | 15.7 | - | - | - | - | - | 16.5 | 33.5 |
| TimeChat (Ren et al., 2024) | ViT-G/14 + LLaMA-2-7B | 47.7 | 22.9 | 12.5 | 30.6 | 30.2 | 16.9 | 8.2 | 21.8 | 14.5 | 23.9 |
| VTimeLLM (Huang et al., 2024a) | CLIP ViT-L/14 + Vicuna-13B | 55.3 | 34.3 | 14.7 | 34.6 | 44.8 | 29.5 | 14.2 | 31.4 | - | - |
| Momentor (Qian et al., 2024) | CLIP ViT-L4 + LLaMA-7B | 42.9 | 23.0 | 12.4 | 29.3 | 42.6 | 26.6 | 11.6 | 28.5 | 7.6 | - |
| HawkEye (Wang et al., 2024b) | - | 50.6 | 31.4 | 14.5 | 33.7 | 49.1 | 29.3 | 10.7 | 32.7 | - | - |
| TRACE (Guo et al., 2025b) | CLIP ViT-L + Mistral-7B | - | 40.3 | 19.4 | - | - | - | - | - | 26.8 | 42.7 |
| NumPro (Wu et al., 2025) | LongVA-7B-DPO | 63.8 | 42.0 | 20.6 | 41.4 | 55.6 | 37.5 | 20.6 | 38.8 | 25.0 | 37.2 |
| *Qwen3-VL-8B-Instruct backbone* | | | | | | | | | | | |
| Qwen3-VL-8B (Bai et al., 2025a) | Qwen3-VL-8B-Instruct | 65.4 | 37.7 | 15.9 | 40.4 | 42.8 | 27.8 | 17.3 | 32.2 | 21.3 | 32.6 |
| *+FT* | Qwen3-VL-8B-Instruct | 65.8 | 44.7 | 21.6 | 42.9 | 58.4 | 39.2 | 21.7 | 40.8 | 24.6 | 36.8 |
| *+F2G-FT(Ours)* | Qwen3-VL-8B-Instruct | **71.3** | **50.4** | **25.7** | **47.2** | **63.8** | **46.1** | **28.4** | **45.7** | **29.7** | **45.6** |

*Table 2.* **Effectiveness of Foresee-to-Ground across various Video-LLMs.**

| Model | Charades-STA | | ActivityNet-Captions | |
|---|---|---|---|---|
| | R@0.7 | mIoU | R@0.7 | mIoU |
| LLaVA-NeXT-7B (Li et al., 2025c) | 7.9 | 24.6 | 6.5 | 12.6 |
| *+F2G-FT* | 12.5 (+4.6) | 32.2 (+7.6) | 12.4 (+5.9) | 23.3 (+10.7) |
| Qwen2.5-VL-7B (Bai et al., 2025b) | 10.3 | 33.6 | 14.4 | 25.7 |
| *+F2G-FT* | 14.2 (+3.9) | 38.6 (+5.0) | 16.7 (+2.3) | 33.5 (+7.8) |
| Qwen3-VL-8B (Bai et al., 2025a) | 15.9 | 40.4 | 17.3 | 32.2 |
| *+FT* | 21.6 | 42.9 | 21.7 | 40.8 |
| *+F2G-FT* | 25.7 (+9.8) | 47.2 (+6.8) | 28.4 (+11.1) | 45.7 (+13.4) |

Overall, explicit evidence citation stabilizes event identification, and evidence-driven measurement sharpens boundary localization. To probe the quality of the video-wide candidate pool, we visualize the Top-$K$ proposal's intervals in Appendix F.1.

**Stability and Robustness.** A key motivation of Foresee-to-Ground is to mitigate the stochasticity of direct timestamp decoding in LLMs. On ActivityNet-Captions, we decode each query twice independently and measure (i) the mean IoU across runs, and (ii) $|\Delta\text{IoU}|$, the run-to-run deviation. As shown in Fig. 4, *F2G* shifts the mean IoU distribution rightward while concentrating substantially more mass near $|\Delta\text{IoU}| = 0$ than the direct timestamp decoding method

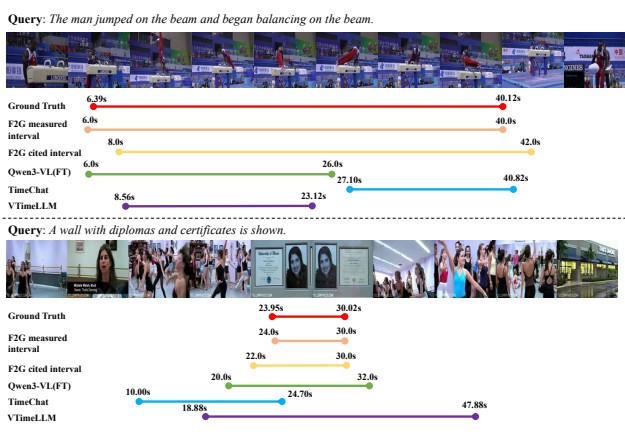

*Figure 3.* **Analysis on special cases.**

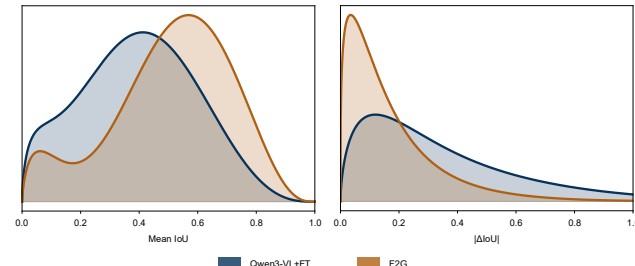

*Figure 4.* **Repeated-inference stability on ActivityNet-Captions.** For visualization, we omit $|\Delta\text{IoU}|$, IoU $\in [0, 0.02)$ and renormalize the remaining density to highlight tail behavior.

(*Qwen3-VL + FT*), indicating higher accuracy with lower variance. We provide an additional failure decomposition analysis in Appendix F.2.

**Efficiency Overhead.** F2G adds a compact evidence interface with modest compute and context cost. Serializing Top-$K$ candidates increases the LLM context by only $\sim$100–200 tokens, which is negligible compared with the original video-token stream (typically on the order of $10^3$–$10^4$ tokens, depending on the sampling rate and visual-token granularity). The added perception components (temporal module, proposal head, and SEE) introduce $\sim$0.5B extra parameters on top of an 8B backbone, and increase end-to-end inference latency by $< 5\%$ under identical decoding settings.

### 4.3 Ablation Studies

**Predictive Temporal Perception.** Table 4 shows that Stage-1 predictive pretraining is critical for high-precision localization. Removing SIGReg or skipping Stage-1 consistently degrades performance, with larger drops at the strict IoU threshold (R@0.7), indicating that dynamics-aware pretraining mainly improves boundary fidelity rather than coarse coverage. Replacing Stage-1 with an action-only Stage-2 recovers only part of the gain, suggesting that generic action supervision cannot fully substitute for the transition cues needed to propose event-like segments.

*Table 3.* Full results on TimeLens-Bench. The best and second-best results in each column are highlighted in bold and underline, respectively.

| Model | Charades-TimeLens | | | | ActivityNet-TimeLens | | | | QVHighlights-TimeLens | | | |
|---|---|---|---|---|---|---|---|---|---|---|---|---|
| | R@0.3 | R@0.5 | R@0.7 | mIoU | R@0.3 | R@0.5 | R@0.7 | mIoU | R@0.3 | R@0.5 | R@0.7 | mIoU |
| Time-R1-7B (Wang et al., 2025b) | 57.9 | 32.0 | 16.9 | 36.6 | 44.8 | 31.0 | 19.0 | 33.1 | 65.8 | 51.5 | 36.1 | 49.2 |
| MiMo-VL-7B (Core Team et al., 2025) | 57.9 | 42.6 | 20.5 | 39.6 | 49.3 | 38.7 | 22.4 | 35.5 | 57.1 | 42.6 | 28.4 | 41.5 |
| Qwen2.5-VL-7B (Bai et al., 2025b) | 59.7 | 37.8 | 16.6 | 39.3 | 44.1 | 31.0 | 16.1 | 31.4 | 41.5 | 27.8 | 15.2 | 31.6 |
| TimeLens-7B (Zhang et al., 2025a) | 70.5 | 55.6 | 28.4 | 48.8 | 62.8 | 51.0 | 32.6 | 46.2 | 74.1 | 62.7 | 43.1 | 56.0 |
| Qwen3-VL-8B (Bai et al., 2025b) | 69.2 | 53.4 | 27.5 | 48.3 | 62.1 | 51.2 | 34.4 | 46.8 | 74.2 | 64.6 | 49.3 | 59.4 |
| TimeLens-8B (Zhang et al., 2025a) | **76.6** | 63.0 | 35.2 | **55.2** | 68.9 | 58.4 | 40.6 | 53.2 | **80.2** | **71.6** | 55.5 | 65.5 |
| **F2G** | 74.2 | **63.1** | **36.3** | 54.9 | **69.8** | **59.4** | **42.3** | **54.1** | 79.6 | 70.4 | 54.5 | 64.1 |

*Table 4.* **Ablations on predictive temporal perception and proposal warm-up (Stage-1/2).**

| Variant | Charades-STA[†] | | ActivityNet-Captions | |
|---|---|---|---|---|
| | R@0.7↑ | mIoU↑ | R@0.7↑ | mIoU↑ |
| **F2G** | 25.7 | 47.2 | 28.4 | 45.7 |
| w/o SIGReg (Stage-1) | 24.1 | 45.8 | 26.8 | 44.2 |
| Random init (no Stage-1) | 20.9 | 43.5 | 22.2 | 41.8 |
| Stage-2 action-only | 21.5 | 44.3 | 24.6 | 43.0 |

**Variant.** *Stage-2 action-only*: skip Stage-1 and train the proposal head in Stage-2 with an action-detection dataset before VTG fine-tuning.

*Table 5.* **Ablations on evidence-driven Identify-then-Measure fine-tuning (Stage-3).**

| Variant | Charades-STA[†] | | ActivityNet-Captions | |
|---|---|---|---|---|
| | R@0.7↑ | mIoU↑ | R@0.7↑ | mIoU↑ |
| **F2G** | 25.7 | 47.2 | 28.4 | 45.7 |
| *Identify / evidence design* | | | | |
| w/o Identify (no ID citation) | 21.5 | 43.1 | 22.2 | 41.1 |
| Interval-only evidence (no $P_k$) | 22.1 | 43.2 | 23.5 | 41.5 |
| *Evidence budget* | | | | |
| Top-$K$=4 (SEE= 4) | 23.3 | 43.5 | 25.1 | 42.2 |
| SEE= 2 (Top-$K$=8) | 24.8 | 45.8 | 27.5 | 44.7 |
| SEE= 8 (Top-$K$=8) | 25.1 | 44.8 | 28.1 | 45.2 |

**Variants.** *w/o Identify* removes the ID-citation requirement and directly generates timestamps. *Interval-only evidence* keeps $(\langle \text{Span}_k \rangle, T_k)$ but removes visual evidence tokens $P_k$. Default uses Top-$K$=8 and SEE= 4.

**Identify-then-Measure Fine-tuning.** Table 5 isolates the key ingredients of evidence-driven Identify-then-Measure fine-tuning. Removing explicit identification (no ID citation) causes a large drop, showing that committing to a discrete candidate segment is an effective constraint for stable and precise boundary generation. Keeping candidate intervals but removing visual evidence tokens further hurts accuracy, indicating that $P_k$ contributes beyond coarse temporal priors by providing discriminative segment content. Finally, the evidence-budget ablations show that reducing Top-$K$ or SEE queries lowers strict-threshold accuracy, while increasing SEE beyond the default gives diminishing returns, supporting Top-$K$=8 and SEE= 4 as a practical operating point.

Viewed together, Tables 4 and 5 separate the contribution of perception from that of structured reasoning. The former improves the quality of the evidence pool, while the latter

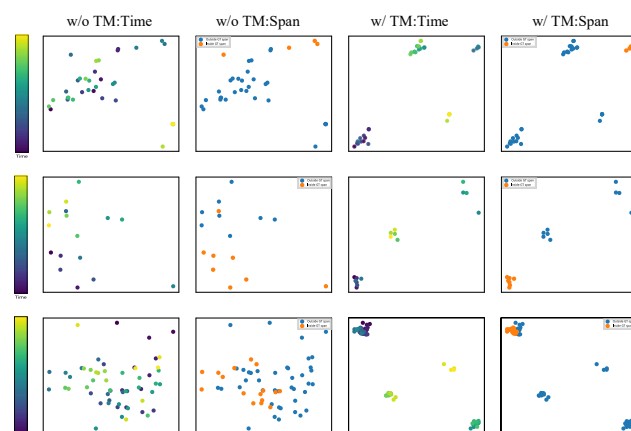

*Figure 5.* **PCA visualization of temporal representations. w/o TM**: without TM, uses temporally pooled visual features, **w/ TM**: with TM, uses temporal module output features. **Time** colors points by temporal order, and **Span** colors points by ground-truth membership (orange: inside; blue: outside). Prediction-trained temporal module makes the embedding trajectory more temporally coherent and increases inside/outside separability, exposing clearer event structure for segment proposal.

determines how the LLM commits to and refines a cited hypothesis. Thus, F2G's gains arise from the interaction between stronger event hypotheses and a supervised, verifiable commitment mechanism, not from temporal features alone.

### 4.4 Mechanism Analysis

**Predictive temporal perception.** To examine how predictive temporal perception reshapes the representation space and captures event changes, we visualize the temporal geometry without and with the temporal module (TM). For each video, we extract a per-timestep temporal sequence from (i) temporally pooled visual features and (ii) TM output features, and project them to 2D via PCA. Compared with the w/o TM features, the prediction-trained TM reshapes the temporal trajectory into a small number of compact clusters that are consistent with the video's coarse event phases. This emergent cluster structure makes event-like segments explicit in representation space, which directly eases span enumeration and boundary-sensitive scoring when constructing the video-wide proposal pool.

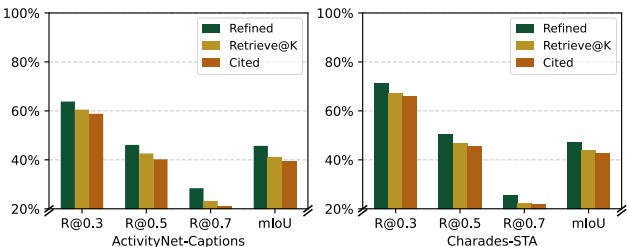

*Figure 6.* **Stage-wise diagnostics for evidence-driven grounding.**

*Table 6.* **Evaluation on VideoMME (w/o subtitles).**

| Model | Overall (%) | Short (%) | Medium (%) | Long (%) |
|---|---|---|---|---|
| Qwen3-VL-8B | 71.4 | 78.5 | 70.1 | 62.3 |
| F2G | 70.9 | 78.3 | 70.2 | 62.1 |

**Evidence-driven Identify-then-Measure.** We further probe the Identify-then-Measure mechanism by decomposing performance into three quantities: *Retrieve@K*, the best span achievable from the Top-$K$ proposal pool without refinement; *Cited*, the evidence segment selected by the model; and *Refined*, the final interval after evidence-conditioned measurement.

Fig. 6 shows a consistent pattern on both ActivityNet-Captions and Charades-STA: *Retrieve@K* provides a tight cite-only upper bound, *Cited* closely tracks this bound, and *Refined* further improves over *Cited*. Thus, the proposal pool provides a coverage-oriented hypothesis space, but the improvement does not come from proposal generation alone: supervised citation commits the LLM to a strong hypothesis, and evidence-conditioned measurement further refines its boundaries. To separate candidate-set coverage from evidence mis-selection, we analyze the per-query citation gap in Appendix F.3.

### 4.5 Preserving General Video Understanding

While Foresee-to-Ground is tailored for temporal grounding, a natural concern is whether VTG-oriented fine-tuning may compromise a backbone's general video understanding. We therefore evaluate Qwen3-VL-8B and F2G (its *+F2G-FT* counterpart) on VideoMME (Fu et al., 2025), a standard benchmark for general video question answering. As shown in Table 6, F2G matches the Qwen3-VL-8B baseline under the official VideoMME metrics, with no observable degradation. These results suggest that F2G improves VTG while preserving broad video QA capability, alleviating concerns about over-specialization.

### 5 Limitations and Future Work

F2G focuses on standard visual VTG, where each query is grounded to a primary temporal interval. This clean setting validates the core Identify-then-Measure formulation, but does not cover the full complexity of real-world video grounding. A natural next step is to extend F2G to complex queries involving multi-event reasoning, compositional temporal constraints, and multiple target spans through hierarchical evidence selection and multi-span Identify-then-Measure. Another promising direction is spatio-temporal grounding, where the evidence unit includes not only a temporal interval but also object-level or region-level support. Finally, incorporating speech, sound events, and environmental audio would allow F2G to move from visual VTG toward audio-visual multimodal grounding. Across these extensions, the central principle remains the same: make intermediate evidence explicit, citable, and verifiable before fine-grained measurement.

### 6 Conclusion

We proposed Foresee-to-Ground (F2G), a verifiable VTG framework that bridges temporal perception and reasoning through an explicit Identify-then-Measure routine. On the perception side, F2G learns boundary-sensitive temporal features via a multi-view latent predictive training with geometry regularization, enabling a lightweight proposal module to extract a compact, video-wide evidence pool of candidate event segments. On the reasoning side, F2G performs evidence-driven temporal grounding: the LLM first identifies the moment by citing a single evidence ID, and then measures precise metric boundaries conditioned on the cited hypothesis, avoiding brittle one-shot timestamp decoding. Empirically, this approach consistently improves grounding accuracy and stability across various Video-LLM backbones without introducing architectural complexity. Ultimately, F2G offers a generalizable paradigm for long-form video agents, demonstrating that decoupling structured temporal discovery from semantic reasoning is key to achieving reliable and auditable video understanding.

### Acknowledgements

This work was supported by the National Natural Science Foundation of China (No. 62272438 and No. 62502115), the Beijing Natural Science Foundation (No. L25700), the Fundamental Research Funds for the Central Universities (No. E2ET1104), and the Hong Kong Scholars Programme.

### Impact Statement

F2G aims to improve the reliability and verifiability of Video-LLM-based temporal grounding by making intermediate segment evidence explicit and citable. It may benefit auditable video search, educational content navigation, assistive video analysis, and human-in-the-loop media review. Our experiments are conducted on established video benchmarks and instruction data, following their intended research-use settings. When applied to real-world video data, especially in privacy-sensitive contexts, F2G should be used with appropriate data licenses, privacy safeguards, and human oversight.

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

## A    Appendix Overview

In this appendix we present:

- The string-level instruction templates and constrained response format used to expose the evidence pool (Section B).

- Stage-wise training data and implementation details for the three-stage pipeline (Section C), including:
    - stage-wise training data and supervision (Section C.1);
    - a detailed module-level data-flow diagram of F2G (Section C.2).

- Examples of the instruction fine-tuning dataset, including both training and evaluation formats (Section D).

- Additional results not included in the main paper (Section E), including:
    - standalone Stage-2 proposal quality (Section E.1);
    - full results across Video-LLMs (Section E.2).

- Additional analyses and diagnostics (Section F), including:
    - visualization and redundancy discussion of the coverage-oriented Top-$K$ proposal pool (Section F.1);
    - repeated-inference stability decomposition (Section F.2);
    - citation-gap analysis to disentangle proposal coverage vs. mis-selection, together with coverage-efficiency implications (Section F.3).

## B    Instruction Templates and Output Format

**Purpose and scope.**    This appendix specifies the string-level instruction used to expose the video-wide evidence pool to the Video-LLM, together with the constrained output format that makes grounding verifiable. As described in the main text, F2G forms an evidence pool $\mathcal{S}_K(V) = \{(\langle \mathtt{Span}_k \rangle, T_k, P_k)\}_{k=1}^{K}$, where $\langle \mathtt{Span}_k \rangle$ is a discrete and citable identifier, $T_k$ is a coarse metric hypothesis (time range), and $P_k$ provides segment evidence in embedding space. The templates below show how we serialize (i) the video stream, (ii) the Top-$K$ proposals/evidence units, and (iii) the query into a single instruction, and how we standardize the response so that the model must *cite exactly one* evidence ID.

**Evidence-unit serialization.**    We serialize each evidence unit as a human-readable candidate entry, and in the templates, $\{K\}$ denotes the number of evidence units in the pool and $\{m\}$ denotes the number of span-level visual tokens allocated per candidate.

Below we provide three templates that implement this interface: i) a training-time instruction used for instruction tuning, ii) an inference-time instruction used for evaluation and deployment, and iii) the response format that appends a single span-ID citation.

---

**Instruction Template(Train)**

<|im_start|>system
You are a multimodal AI assistant. Follow the user instruction and produce the answer.
<|im_end|>

<|im_start|>user
% Video stream
<0.3 seconds><|vision_start|><|video_pad|><|video_pad|><|video_pad|><|video_pad|><|video_pad|><|video_pad|><|video_pad|><|video_pad|>...<|video_pad|><|vision_end|><1.3 seconds><|vision_start|><|video_pad|>...<|video_pad|><|vision_end|>

% Candidates
Here are {K} candidate event spans extracted from the video. Each candidate provides (1) its time range and (2) {m} visual span tokens. You MUST cite exactly one span id token at the end of your

---

answer.
Candidate {1}: from X seconds to Y seconds, <Span_1> <|vision_start|><|video_pad|><|video_pad|><|vision_end|>
...
Candidate {K}: from X seconds to Y seconds, <Span_{K}> <|vision_start|><|video_pad|><|video_pad|><|vision_end|>

% Question
{question}, Please answer naturally, and finally cite exactly ONE candidate span id token: <Span_1> <Span_2> <Span_3> <Span_4> <Span_5> <Span_6> <Span_7> <Span_8> .
<|im_end|>

---

### Instruction Template(Inference)

<|im_start|>system
You are a multimodal AI assistant. Follow the user instruction and produce the answer.
<|im_end|>

<|im_start|>user
% Video stream
<0.3 seconds><|vision_start|><|video_pad|><|video_pad|><|video_pad|><|video_pad|><|video_pad|><|video_pad|><|video_pad|><|video_pad|>...<|video_pad|><|vision_end|><1.3 seconds><|vision_start|><|video_pad|>...<|video_pad|><|vision_end|>

% Candidates
Here are {K} candidate event spans extracted from the video. Each candidate provides (1) its time range and (2) {m} visual span tokens. You MUST cite exactly one span id token at the end of your answer.
Candidate {1}: from X seconds to Y seconds, <Span_1> <|vision_start|><|video_pad|><|video_pad|><|vision_end|>
...
Candidate {K}: from X seconds to Y seconds, <Span_{K}> <|vision_start|><|video_pad|><|video_pad|><|vision_end|>

% Query
During what time, can you see {Query}. Please answer naturally, and finally cite exactly ONE candidate span id token: <Span_1> <Span_2> <Span_3> <Span_4> <Span_5> <Span_6> <Span_7> <Span_8> .
<|im_end|>

---

### Response Format

<|im_end|>
{answer}. Corresponding span: <Span_PLACEHOLDER>.
<|im_end|>

*Table 7.* **Stage-wise training data and supervision.**

| Stage | Data | Supervision | Trainable components |
|---|---|---|---|
| Stage-1 | ∼200K unlabeled clips | $\mathcal{L}_{\text{pred}} + \mathcal{L}_{\text{SIG}}$ | $f_\theta, g_\phi$ |
| Stage-2 | ∼10K interval-annotated videos | regression + objectness scoring | $f_\theta, h_\psi$ |
| Stage-3 | 220K VTG instructions | LM + ID + time losses | LoRA, SEE, $f_\theta/h_\psi$ with small LR |

## C  Stage-wise Training Data and Implementation Details

### C.1  Stage-wise Training Data and Supervision

Table 7 summarizes the data, supervision, and trainable components used in the three-stage training pipeline.

*Stage-1.* Self-supervised predictive temporal pretraining on ∼200K unlabeled video clips.

*Stage-2.* Query-agnostic proposal warm-up on ∼10K interval-annotated videos.

*Stage-3.* Evidence-driven Identify-then-Measure adaptation on 220K VTG instruction samples.

### C.2  Detailed Data Flow

Fig. 7 provides a module-level data-flow diagram of F2G across the three training stages. It complements Fig. 2 by expanding the computation inside the Temporal Module, Proposal Head, and Span Evidence Encoder.

## D  Instruction fine-tuning dataset

**Overview.**  This section provides concrete examples of the instruction fine-tuning data used in F2G. The training split follows an instruction-style format, where QA pairs may cover temporal localization as well as broader video-understanding intents, which regularizes learning and mitigates overfitting to a single VTG phrasing. In contrast, evaluation follows the standard VTG protocol by providing the query and annotated start/end times for metric scoring.

```
Train Dataset

{
    "id": "v_QOlSCBRmfWY_q1",
    "source": "anet",
    "video": "v_QOlSCBRmfWY",
    "duration": 82.7,
    "question": "Can you provide more details about the dance itself?",
    "answer": "Certainly! The young woman engages in a lively dance, moving around the room from
18.0 seconds to 60.0 seconds. Her movements showcase a range of emotions and expressions as she
dances. The dance concludes with her lying on the floor from 56.0 seconds to 80.0 seconds."
 },
```

```
Evaluation Dataset

{
    "id": "v_uqiMw7tQ1Cc_1",
    "video": "v_uqiMw7tQ1Cc.mp4",
    "start_time": 0,
    "end_time": 4.14,
    "query": "Two men both dressed in athletic gear are standing and talking in an indoor weight
lifting gym filled with other equipment.",
    "duration": 55.15
 },
```

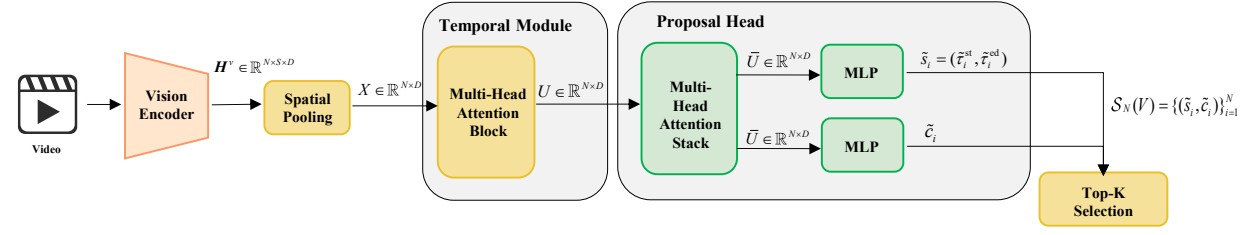

Stage1: Predictive Temporal Perception Pretraining

Stage2: Proposal Warm-Up

Stage3: Identify-then-Measure Fine-Tuning

*Figure 7.* Detailed data flow of additional modules across the three training stages.

## E   Additional Results

### E.1   Standalone Stage-2 Proposal Quality

To evaluate the proposal head independently of downstream LLM reasoning, we report Stage-2-only metrics before Top-$K$ evidence serialization. The proposal head produces dense per-timestep predictions, including an eventness score and a regressed temporal span. We report Center-AP to measure event-center classification quality and Matched mIoU to measure span localization quality after matching predicted spans to ground-truth intervals.

As shown in Table 8, the proposal head learns a meaningful proposal space before evidence-driven LLM fine-tuning. These results characterize the independent quality of Stage-2 proposal generation, while the final grounding performance further depends on evidence citation and boundary refinement in Stage-3.

### E.2   Full Results Across Video-LLMs

Table 9 reports the complete per-metric results for the cross-backbone study. For each Video-LLM, *+F2G-FT* applies the same fine-tuning recipe; the subscript $(+\cdot)$ denotes the absolute improvement over the untuned backbone in the same row group. For Qwen3-VL-8B, we additionally include *+FT* to separate gains from conventional instruction fine-tuning.

*Table 8.* **Standalone proposal quality after Stage-2 warm-up.**

| Metric | Value |
|---|---|
| Center-AP | 0.79 |
| Matched mIoU | 0.56 |

*Table 9.* **Effectiveness of Foresee-to-Ground across various Video-LLMs (Full results).**

| Model | Charades-STA | | | | ActivityNet-Captions | | | |
|---|---|---|---|---|---|---|---|---|
| | R@0.3 | R@0.5 | R@0.7 | mIoU | R@0.3 | R@0.5 | R@0.7 | mIoU |
| LLaVA-NeXT-7B (Li et al., 2025c) | 30.1 | 13.8 | 7.9 | 24.6 | 18.6 | 9.8 | 6.5 | 12.6 |
| +F2G-FT | 43.2$_{(+13.1)}$ | 26.2$_{(+12.4)}$ | 12.5$_{(+4.6)}$ | 32.2$_{(+7.6)}$ | 29.5$_{(+10.9)}$ | 19.8$_{(+10.0)}$ | 12.4$_{(+5.9)}$ | 23.3$_{(+10.7)}$ |
| Qwen2.5-VL-7B (Bai et al., 2025b) | 51.1 | 29.5 | 10.3 | 33.6 | 33.4 | 22.3 | 14.4 | 25.7 |
| +F2G-FT | 62.5$_{(+11.4)}$ | 36.8$_{(+7.3)}$ | 14.2$_{(+3.9)}$ | 38.6$_{(+5.0)}$ | 46.7$_{(+13.3)}$ | 32.6$_{(+10.3)}$ | 16.7$_{(+2.3)}$ | 33.5$_{(+7.8)}$ |
| Qwen3-VL-8B (Bai et al., 2025a) | 65.4 | 37.7 | 15.9 | 40.4 | 42.8 | 27.8 | 17.3 | 32.2 |
| +FT | 65.8 | 44.7 | 21.6 | 42.9 | 58.4 | 39.2 | 21.7 | 40.8 |
| +F2G-FT | 71.3$_{(+5.9)}$ | 50.4$_{(+12.7)}$ | 25.7$_{(+9.8)}$ | 47.2$_{(+6.8)}$ | 63.8$_{(+21.0)}$ | 46.1$_{(+18.3)}$ | 28.4$_{(+11.1)}$ | 45.7$_{(+13.4)}$ |

# F Additional Analyses

## F.1 Coverage-Oriented Top-$K$ Proposal Pool Visualization

To complement the case-study comparisons in Sec. 4.2, we visualize the raw Top-$K$ candidate segments produced by the proposal head.

Fig. 8 shows two examples. Each row corresponds to one video-query pair, with the thumbnail strip indicating the temporal context and the colored bars showing the Top-$K$ proposed intervals in seconds. The proposal pool is designed to be coverage-oriented under a small evidence budget: it aims to preserve plausible event hypotheses for subsequent evidence citation and measurement, rather than to produce a maximally non-overlapping set of intervals. Accordingly, the pool may contain both tight segments and longer spans, as well as locally overlapping proposals. Such overlap can correspond to alternative boundary hypotheses around the same semantic event, rather than pure redundancy.

However, excessive overlap may reduce the effective coverage of long or event-dense videos, which motivates the coverage-efficiency analysis in Sec. F.3.

## F.2 Additional Stability Diagnostics

Beyond the distributional evidence in Fig. 4, we further analyze how repeated-inference variance occurs. For each query in ActivityNet-Captions, we decode twice independently and obtain two IoU scores, $\text{IoU}_1$ and $\text{IoU}_2$, with respect to the ground-truth interval. We focus on degenerate failures at $\text{IoU} = 0$ and divide them into two cases: *consistent-miss*, where $\text{IoU}_1 = \text{IoU}_2 = 0$, and *stochastic-collapse*, where exactly one run yields zero IoU: $[\text{IoU}_1 = 0] \oplus [\text{IoU}_2 = 0]$. For stochastic-collapse cases, we further stratify examples by the non-zero IoU of the other run to measure collapse severity.

As shown in Fig. 9, direct timestamp decoding exhibits a noticeable fraction of stochastic-collapse cases, meaning that identical inputs can lead to success or failure depending on sampling noise. In contrast, *F2G* substantially reduces such collapse, consistent with its explicit evidence commitment. The remaining errors are dominated by consistent-miss cases, which more likely reflect intrinsically difficult queries, such as severe ambiguity or weak visual evidence. This complements Fig. 4 by showing that *F2G* improves stability mainly by suppressing sampling-induced collapse.

## F.3 Citation-gap analysis

A natural follow-up to the stage-wise diagnostics in Fig. 6 is to identify whether remaining errors come from insufficient proposal coverage or suboptimal evidence selection. To disentangle these two sources, we analyze the citation gap between the best candidate available in the proposal pool and the span actually cited by the model.

For each query with ground-truth interval $T^\star$, we compute the best IoU achievable within the Top-$K$ proposal pool,

$$\text{IoU}_{\text{best}} = \max_{k \in \{1, \ldots, K\}} \text{IoU}(T_k, T^\star), \tag{20}$$

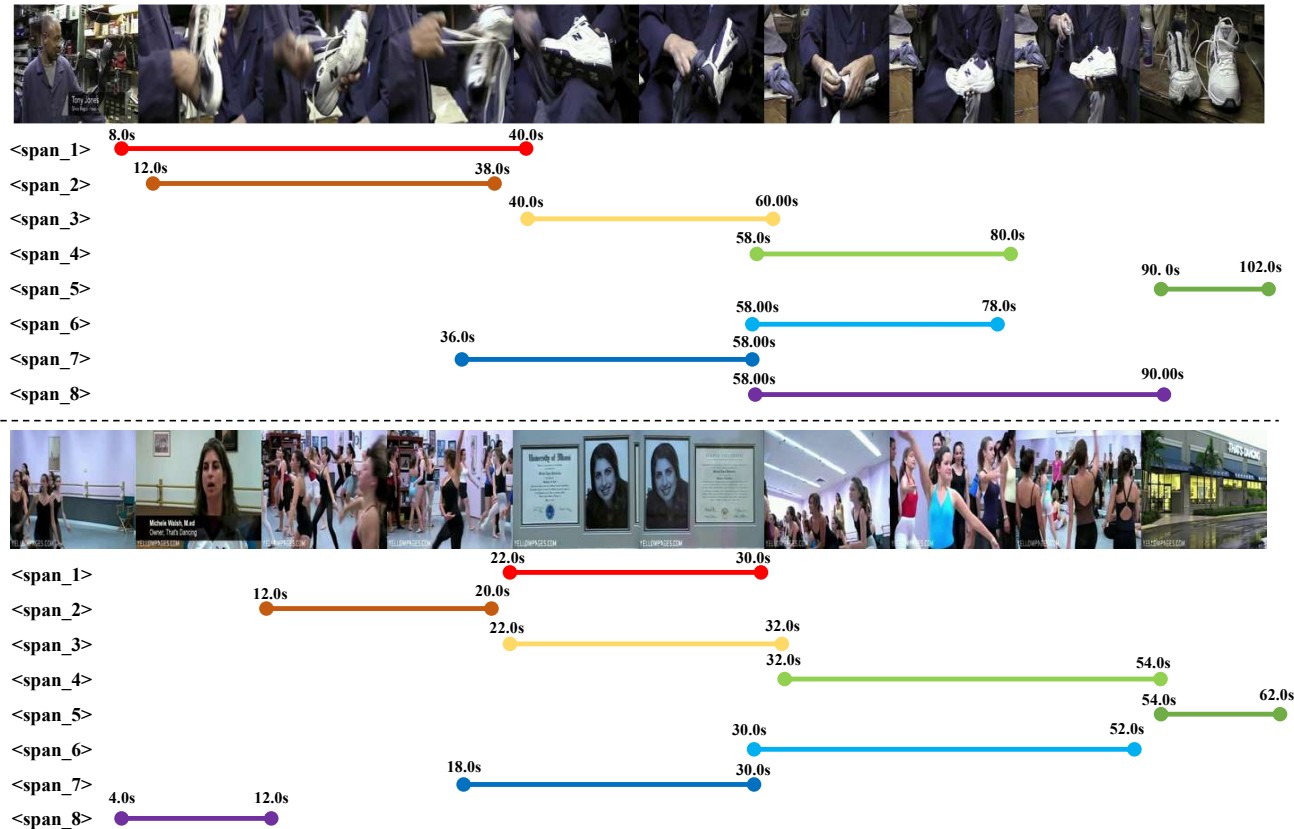

*Figure 8.* **Visualization of the Top-$K$ proposal pool.** For each video pair, we plot the Top-$K$ candidate segments $T_k$ (in seconds), ordered by proposal objectness (top to bottom). The resulting pool provides diverse, video-wide event hypotheses that are later serialized as evidence units for Identify→Measure grounding.

and the IoU of the cited span,

$$\text{IoU}_{\text{cite}} = \text{IoU}(T_z, T^\star), \tag{21}$$

where $z$ is the model-cited index. We then define the citation gap

$$\Delta\text{IoU} = \text{IoU}_{\text{best}} - \text{IoU}_{\text{cite}} \in [0, 1]. \tag{22}$$

A small $\Delta\text{IoU}$ indicates that the model's citation is near-optimal *given the pool*, whereas a large gap indicates that errors are attributable to mis-selection rather than pool quality.

Fig. 10 plots the distribution of $\Delta\text{IoU}$. The mass concentrates near zero on both ActivityNet-Captions and Charades-STA: 87.8% and 93.6% of queries have $\Delta\text{IoU} < 0.10$, respectively. This shows that, once a good candidate exists in the pool, the Identify step usually selects a near-optimal span. Thus, remaining errors are more often limited by candidate-set coverage, i.e., $\text{IoU}_{\text{best}}$ itself being low, rather than by evidence mis-selection. Together with Fig. 6, this supports the view that improving proposal quality and coverage efficiency is the main lever for further gains, while evidence-conditioned measurement sharpens boundaries beyond the cited hypothesis.

This also clarifies the redundancy–coverage trade-off in Sec. F.1: under a fixed Top-$K$ budget, moderate local overlap can preserve alternative boundary hypotheses, whereas excessive overlap may reduce effective coverage. Future extensions may improve coverage efficiency through multi-scale proposal generation, diversity-aware temporal suppression, adaptive Top-$K$ allocation, or coarse-to-local proposal refinement.

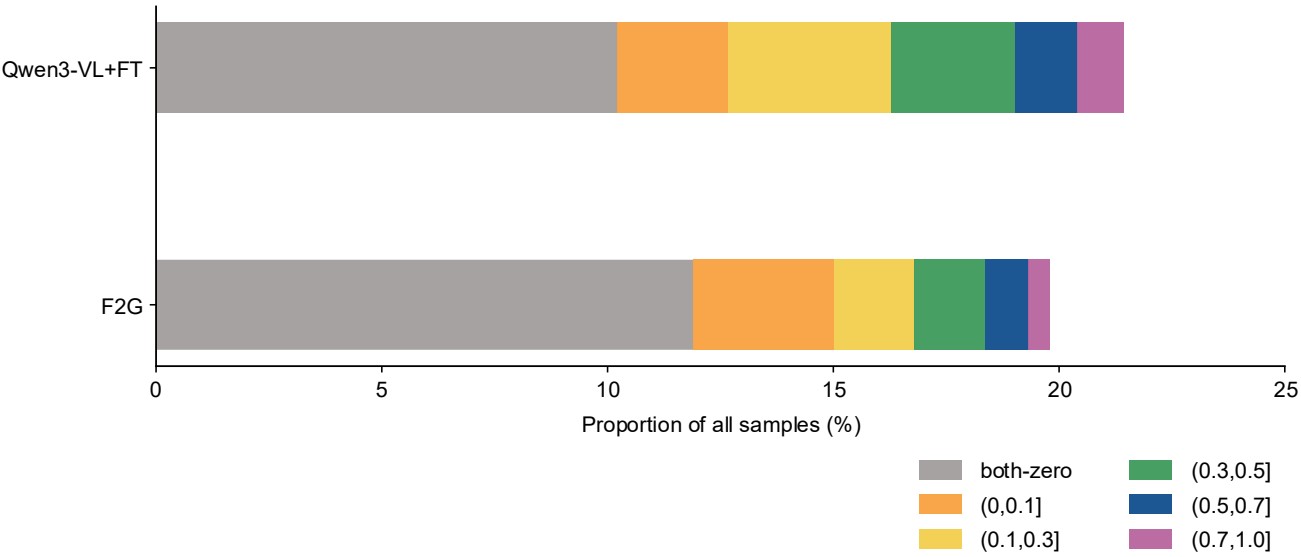

*Figure 9.* **Repeated-inference failure decomposition on ActivityNet-Captions.** We decompose repeated decoding outcomes into consistent-miss ($\text{IoU}_1 = \text{IoU}_2 = 0$) and stochastic-collapse (exactly one run has $\text{IoU} = 0$), and further stratify collapse cases by the other run's non-zero IoU.

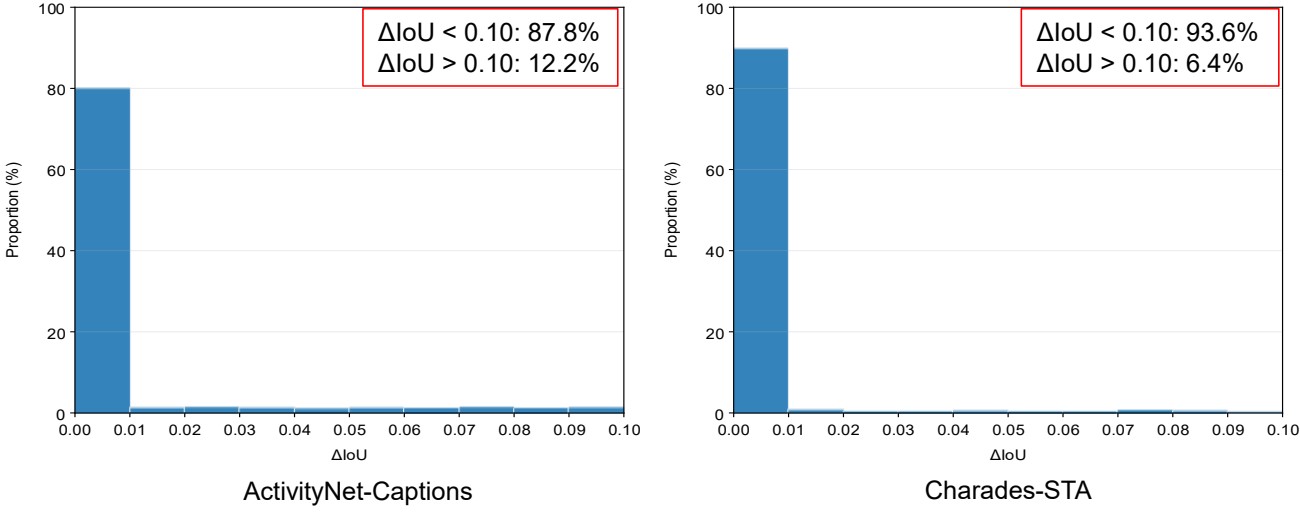

*Figure 10.* **Citation-gap distribution.** We plot $\Delta\text{IoU} = \text{IoU}_{\text{best}} - \text{IoU}_{\text{cite}}$, where $\text{IoU}_{\text{best}}$ is the best candidate IoU within the Top-$K$ proposal pool and $\text{IoU}_{\text{cite}}$ is the IoU of the model-cited span. Most mass lies near $\Delta\text{IoU} = 0$ (87.8% / 93.6% of queries have $\Delta\text{IoU} < 0.10$ on ActivityNet-Captions / Charades-STA), indicating that citation is usually near-optimal given the pool; remaining failures are therefore more often constrained by candidate-set coverage than by mis-selection.

