# OpenReview forum: "Foresee-to-Ground: From Predictive Temporal Perception to Evidence-Driven Reasoning for Video Temporal Grounding"
_ICML.cc/2026/Conference — ICML 2026 regular_

### Official Review · Reviewer_ngwV · 2026-02-14

**Soundness:** 3
**Presentation:** 2
**Significance:** 2
**Originality:** 3
**Overall Recommendation:** 4
**Confidence:** 3

**Summary:**

This paper introduces Foresee-to-Ground (F2G), a VTG framework that formulates the task as a verifiable Identify-then-Measure pipeline. Addtional temporal modules are proposed to extract event features and generate temporal proposals. Extensive experimental results on three datasets demonstrate the effectiveness.

**Compliance With Llm Reviewing Policy:**

Affirmed.

**Final Justification:**

Thanks for the authors' response, which addresses most of my concerns. I hope these modifications will be included in the final revision. I tend to keep my original rating.

**Key Questions For Authors:**

1. While the paper claims that this is a decoupled method, first identifying <Span_k> token and then predicting timestamps conditioned on P_k and T_k, the method actually feed all P_i and T_i into the LLM for final answer generation. Moreover, the LLM output is a single-pass response, which directly predicts the answer and finally provides the span token. These seem to contradict the claimed "Identify-then-Measure" formulation.
2. A cross-comparison of Table 1 and Figure 6 suggests that the temporal proposals alone already surpasses the existing SOTA methods. This raises my concerns: does the main contribution lies in the stronger proposal generator rather than the proposed "Identify-then-Measure"decoupled formulation?
3. Lacking details about how $X_g$ and $X_l$ derived and how $N_g$ / $N_v$ determined.
4. It would be beneficial to provide a more detailed data flow diagram to illustrate the computation within the three additional modules.

**Limitations:**

yes

**Strengths And Weaknesses:**

1. The combination of discrete and continuous representations are interesting.
2. The paper is well-written.
3. Extensive ablation studies and analysis are presented.

---

> ### Author Rebuttal · Authors · 2026-03-31
>
> Thank you for the careful reading and constructive feedback. We appreciate your positive assessment of the paper’s presentation and empirical analysis. Your main concerns about the Identify-then-Measure formulation, the respective roles of proposal and reasoning, and the clarity of several implementation details are very helpful, and we address them below.
>
> **1. On whether F2G is truly “Identify-then-Measure.”**
>
> Our claim is **not** that F2G requires two physically separate LLM passes; the decoupling is at the **formulation and supervision level**. We enforce **Identify-then-Measure** through the prompt and constrained output format (Appendix B): the LLM is instructed to cite exactly one span ID from the evidence pool and then generate the final timestamp prediction under that cited hypothesis. In this sense, **Identify** is explicit hypothesis selection from the evidence pool, and **Measure** is temporal grounding conditioned on that cited hypothesis. Therefore, the key property is explicit hypothesis selection with structured supervision on both the cited span ID and the timestamp output, rather than whether the citation token appears earlier in the serialized output string. We will revise the wording to make clear that F2G is a structured factorization within one response, rather than a mandatory two-pass execution.
>
> **2. On whether the main contribution is the proposal generator.**
>
> We agree that the proposal module is a major component and largely determines the achievable upper bound. However, proposal generation alone does not complete VTG: it provides a high-recall candidate space, while the final task still requires selecting the most relevant hypothesis and refining it into a precise and stable prediction. This is why we analyze **Retrieve@K / Cited / Refined** (Fig. 6) separately: proposal coverage defines the candidate upper bound, citation tests whether the right hypothesis is identified, and evidence-conditioned refinement sharpens the final boundary. Proposal quality and Identify-then-Measure are therefore complementary rather than interchangeable. This is also supported by Table 4, where removing explicit **Identify** causes a substantial drop. We will make this division of roles more explicit in the revision.
>
> **3. On the definitions of $X_g$, $X_l$, $N_g$, and $N_v$.**
>
> Eq. (3) first defines the full temporal sequence $X = \mathrm{Pool}(H^v)$. In Stage-1 multi-view pretraining, $X_g$ and $\{X_l^{(v)}\}$ are sampled from this same sequence: $X_g$ is the global temporal view, while $\{X_l^{(v)}\}$ are local temporal views from partial observations such as shorter crops or strided subsequences. Correspondingly, $N_g$ and $N_v$ denote the timestep counts of the global and local-view sequences. After Stage-1, we discard the predictor and return to the standard full-sequence path $U=f_\theta(X)$ for Stage-2/3. We will revise this transition more explicitly.
>
> **4. On the request for a more detailed data-flow diagram.**
>
> We have prepared a module-level diagram showing how features flow through the Temporal Module, Proposal Head, and Span Evidence Encoder across the three training stages. The anonymous link is: [Data-flow Diagram](https://anonymous.4open.science/r/Data-flow-diagram). We will include this diagram in the revised appendix.
>
> Thank you again for the helpful feedback. We will revise the paper to clarify the formulation-level decoupling in Identify-then-Measure, better separate the roles of proposal and reasoning, and improve the presentation of notation and data flow.

---

> > ### Author Rebuttal · Reviewer_ngwV · 2026-04-03
> >
> > Thanks for the authors' response, which addresses most of my concerns. I hope these modifications will be included in the final revision. I tend to keep my original rating.

---

> > > ### Author Response · Authors · 2026-04-04
> > >
> > > Thank you for the update and for confirming that our response addressed your concerns. We will make sure these clarifications and modifications are included in the final revision. We appreciate your time and consideration.

---

### Official Review · Reviewer_XqQZ · 2026-03-05

**Soundness:** 3
**Presentation:** 3
**Significance:** 3
**Originality:** 3
**Overall Recommendation:** 4
**Confidence:** 4

**Summary:**

This paper targets the core limitations of Video Large Language Model (Video-LLM) based Video Temporal Grounding (VTG). Existing methods treat VTG as direct timestamp regression, which creates a fundamental misalignment between LLMs' discrete token space and continuous temporal coordinates, leading to brittle predictions, inconsistent boundaries, and high inference variance.
To address this, the authors propose Foresee-to-Ground (F2G), a novel VTG framework built on a verifiable Identify-then-Measure routine aligned with human cognitive logic. F2G consists of two synergistic components: (1) a Predictive Temporal Perception module that learns boundary-sensitive temporal representations via multi-view latent prediction, to construct a compact video-wide evidence pool of candidate event segments; (2) an Evidence-Driven Reasoning module that augments LLM inputs with citable evidence units, forcing the model to first identify the target moment by citing an evidence ID before refining precise temporal boundaries under the cited hypothesis. Extensive experiments show that F2G consistently outperforms state-of-the-art VTG methods across standard benchmarks with particularly large gains at strict IoU thresholds.

**Compliance With Llm Reviewing Policy:**

Affirmed.

**Key Questions For Authors:**

See weakness.

**Limitations:**

See weakness.

**Strengths And Weaknesses:**

### Strengths

1. Module design combines solid theoretical support with ingenious engineering implementation, with its effectiveness fully and rigorously validated. On the perception module side, the Predictive Temporal Perception module is built on the core insight that "the dynamics within an event are predictable, while prediction uncertainty is high at event boundaries". It learns boundary-sensitive temporal representations automatically through a self-supervised objective of multi-view latent space prediction, and introduces the SIGReg regularizer to stabilize the geometric distribution of the latent space. This design is backed by well-established self-supervised learning theories, rather than pure engineering tricks. On the reasoning module side, the design of citable evidence units perfectly adapts to the instruction-following and discrete token modeling capabilities of LLMs, providing structured anchors for LLMs with a minimal context overhead (only 100-200 additional tokens).

2. Meanwhile, the paper comprehensively verifies the necessity and working mechanism of each core module through multiple sets of ablation studies, PCA visualization, and citation gap analysis, forming a complete and self-consistent logical closed loop.


### Weaknesses

1. Lack of validation for scenario and benchmark generalization, and evaluations on TimeLens and Vidi are required to draw more solid conclusions, the existing experiments only cover conventional VTG benchmarks, without validation on challenging benchmarks focusing on fine-grained temporal understanding and complex event localization such as TimeLens[1] and Vidi[2]. Supplementary evaluations on these scenarios and benchmarks are needed to make the conclusions on the effectiveness and generalization of the model more convincing and solid.

2. Due to its lightweight structural design, for sophisticated and composite natural language queries, the evidence selection mechanism presented in this work may be insufficient to provide robust and accurate matching, which limits its performance in complex VTG scenarios.

[1] TimeLens: Rethinking Video Temporal Grounding with Multimodal LLMs

[2] Vidi: Large Multimodal Models for Video Understanding and Editing

---

> ### Author Rebuttal · Authors · 2026-03-31
>
> Thank you for the thoughtful and encouraging feedback. We appreciate your recognition of F2G’s core insight, overall design, and the strength of the ablation and validation results. Your suggestions on more challenging benchmarks and the current limits in complex VTG scenarios are very valuable, and we address them below.
>
> **1. On validation on more challenging benchmarks (TimeLens / Vidi).**
>
> To directly address it, we added an evaluation on **TimeLens** using the same training/evaluation data protocol as the TimeLens paper: we keep **Stage-1/2** unchanged, replace only **Stage-3** training data with **TimeLens-100K**, and evaluate on **TimeLens-Bench**. This directly tests whether F2G transfers to a stricter VTG benchmark.
>
> As shown below, F2G remains highly competitive on TimeLens-Bench. It consistently improves over the corresponding **Qwen3-VL-8B** baseline and reaches performance close to the best reported results, while still using only **pure SFT** without RL. This makes the comparison conservative for F2G, since the final **TimeLens-8B** models use a stronger pipeline with **thinking-free RLVR**. We will add these results in the revision.
>
> **Table 1. Results on TimeLens-Bench.**
> **Bold** = best, *italic* = second best, C = Charades-TimeLens, A = ActivityNet-TimeLens, and Q = QVHighlights-TimeLens.
>
> | Model | C@0.3 | C@0.5 | C@0.7 | C-mIoU | A@0.3 | A@0.5 | A@0.7 | A-mIoU | Q@0.3 | Q@0.5 | Q@0.7 | Q-mIoU |
> |---|---:|---:|---:|---:|---:|---:|---:|---:|---:|---:|---:|---:|
> | Time-R1-7B | 57.9 | 32.0 | 16.9 | 36.6 | 44.8 | 31.0 | 19.0 | 33.1 | 65.8 | 51.5 | 36.1 | 49.2 |
> | MiMo-VL-7B | 57.9 | 42.6 | 20.5 | 39.6 | 49.3 | 38.7 | 22.4 | 35.5 | 57.1 | 42.6 | 28.4 | 41.5 |
> | Qwen2.5-VL-7B | 59.7 | 37.8 | 16.6 | 39.3 | 44.1 | 31.0 | 16.1 | 31.4 | 41.5 | 27.8 | 15.2 | 31.6 |
> | TimeLens-7B | 70.5 | 55.6 | 28.4 | 48.8 | 62.8 | 51.0 | 32.6 | 46.2 | 74.1 | 62.7 | 43.1 | 56.0 |
> | Qwen3-VL-8B | 69.2 | 53.4 | 27.5 | 48.3 | 62.1 | 51.2 | 34.4 | 46.8 | 74.2 | 64.6 | 49.3 | 59.4 |
> | TimeLens-8B | **76.6** | *63.0* | *35.2* | **55.2** | *68.9* | *58.4* | *40.6* | *53.2* | **80.2** | **71.6** | **55.5** | **65.5** |
> | **F2G** | *74.2* | **63.1** | **36.3** | *54.9* | **69.8** | **59.4** | **42.3** | **54.1** | *79.6* | *70.4* | *54.5* | *64.1* |
>
> **Vidi (VUE-TR)** is also an important benchmark. However, it targets a broader setting than our current formulation, including **audio-supported queries** and **multiple retrieved time ranges**, rather than standard visual VTG with a primary target span. We therefore prioritized **TimeLens** in this rebuttal as the more directly aligned benchmark, and view **Vidi** as an important next-step benchmark for future extensions of F2G.
>
> **2. On the current limits in complex VTG scenarios.**
>
> We agree that the current lightweight evidence-selection mechanism may limit performance on more complex VTG scenarios, especially for queries requiring multiple-event reasoning or compositional temporal constraints. Since the current framework is designed for standard VTG, we plan to extend **F2G** to more complex VTG scenarios through **hierarchical evidence selection**, **query-aware reranking / iterative evidence expansion**, **multi-span Identify-then-Measure**, and potentially **spatio-temporal Identify-then-Measure**.
>
> Thank you again for the insightful suggestions. We will revise the paper to include the additional TimeLens validation and to better clarify the current scope of F2G and its future extensions to more complex VTG scenarios.

---

> > ### Author Rebuttal · Reviewer_XqQZ · 2026-04-04
> >
> > Thank the authors for their detailed rebuttal. I maintain my support for acceptance of this paper.

---

> > > ### Author Response · Authors · 2026-04-04
> > >
> > > Thank you very much for the update and for your support. We truly appreciate your careful reading and thoughtful feedback.
> > >
> > > We will make sure the promised additions are included in the revision, especially the TimeLens evaluation and the clarified discussion of F2G’s future extensions.

---

### Official Review · Reviewer_Biuf · 2026-03-08

**Soundness:** 3
**Presentation:** 3
**Significance:** 3
**Originality:** 3
**Overall Recommendation:** 4
**Confidence:** 3

**Summary:**

This paper addresses the Video Temporal Grounding task by proposing Foresee-to-Ground (F2G), a framework that replaces traditional black-box timestamp regression with a verifiable Identify-then-Measure pipeline. F2G operates in three distinct stages: first, it learns boundary-sensitive features through Predictive Temporal Perception; second, it constructs a video-wide evidence pool using a lightweight proposal module; and third, it fine-tunes a Video-LLM to first cite a specific evidence ID before performing local boundary refinement. The method demonstrates stronger performance over standard instruction fine-tuning and exhibits robust stability and generalization across various backbones.

**Compliance With Llm Reviewing Policy:**

Affirmed.

**Final Justification:**

The rebuttal has addressed my concerns. Considering other reviewers' opinions, I hope the analyses in the rebuttal can be included in the final version.

**Key Questions For Authors:**

Please see the weakness part, mainly weakness 1,2,4,5.

If the author can address my concern, I will raise my score.

**Limitations:**

yes.

**Strengths And Weaknesses:**

**Strength**
1. **Effective Predictive Temporal Pretraining**: The Stage-1 multi-view latent predictive pretraining successfully encourages the model to distinguish stable within-event evolution from boundary-induced ambiguity, resulting in boundary-sensitive representations that improve grounding fidelity.
2. **Enhanced Stability and Robustness**: By enforcing an explicit citation of a candidate segment prior to boundary generation, F2G significantly mitigates the stochasticity of direct timestamp decoding, as shown in Figure 4.
3. **Broad Generalization & Efficiency**: F2G shows consistent accuracy gains across multiple Video-LLM backbones and preserves general video QA capabilities. Furthermore, the architectural overhead is minimal, introducing only ~0.5B parameters and less than 5% additional inference latency.

**Weakness**
1. **Proposal Redundancy**: The visualization in Appendix E.2 (Figure 8) reveals that the proposal head often generates highly redundant or overlapping candidate segments. This suggests that the Top-K pool may not be as diverse as intended. Is there any approach to alleviate such issue?
2. **Coverage as a Performance Bottleneck**: According to the citation-gap analysis in Section E.3, the model's citation is usually near-optimal given the pool. This implies that the primary factor limiting the performance upper bound is the coverage of the proposal pool itself rather than the LLM's reasoning. How do the authors plan to improve the recall/coverage of the Stage-2 proposals without significantly increasing K?
3. **Generalization of the Proposal Head**: While the LLM shows zero-shot transfer, the Stage-2 proposal head is trained on specific VTG intervals. There is a concern regarding how well this lightweight head generalizes to out-of-distribution video domains compared to the base LLM's internal perception.
4. **Missing Ablation Studies**: The paper provides a comparison between F2G and standard SFT, but it lacks Stage 1 + standard instructional finetuning. This would clarify whether the gains stem from the improved features or the structural Identify-then-Measure logic.
5. **Incomplete Implementation Details**: The manuscript does not explicitly specify the exact training data used for Stage-1/2/3. Additionally, reporting the standalone performance of the proposal head after Stage-2 would provide a clearer picture of the perception module's independent quality.

---

> ### Author Rebuttal · Authors · 2026-03-31
>
> Thank you for the thoughtful and constructive feedback. We appreciate your positive assessment of predictive temporal pretraining and the explicit citation mechanism. Your comments on proposal redundancy, coverage, generalization, disentangling the gains from perception and reasoning, and implementation clarity are highly valuable. We address them below.
>
> **1. On proposal redundancy.**
>
> Our current Stage-2 design is coverage-first: the goal is to build a high-recall candidate set under a limited budget, rather than maximize diversity itself. Some local redundancy is therefore not necessarily harmful and may even improve downstream boundary refinement. That said, excessive redundancy is more problematic in long or event-dense videos. Promising directions include adaptive Top-$K$ selection, diversity-aware ranking / temporal suppression, and lightweight query-aware reranking.
>
> **2. On proposal coverage as the bottleneck.**
>
> We agree that proposal coverage is a main bottleneck. Our analysis suggests that once the pool contains a sufficiently good hypothesis, citation is usually close to the pool optimum, while the Measure step still improves the final boundary beyond the cited span. We therefore view the issue as coverage efficiency under a limited evidence budget, rather than proposal quality alone. Instead of simply increasing $K$, more promising directions include stronger multi-scale proposal generation, coverage/diversity-aware ranking, adaptive Top-$K$, and coarse global proposal + local refinement.
>
> **3. On the generalization of the proposal head.**
>
> Stage-2 supervision is fundamentally **query-agnostic interval supervision**: its role is not to learn benchmark-specific query-answer patterns, but to warm up proposal formatting, temporal localization, and alignment with the downstream evidence interface. We use VTG-style interval annotations because they are high-quality and readily available, but this supervision is not tied to a specific query ontology and could in principle also come from interval-annotated videos from other domains.
>
> **4. On the suggested disentangling ablation.**
>
> The key question is whether the gains come from stronger temporal perception or from Identify-then-Measure itself. In our pipeline, Stage-1 alone does not interface with the LLM; only after Stage-2 are features converted into the proposal/evidence interface. The closest comparable setting is therefore to keep Stage-1/2 but remove explicit Identify and use standard timestamp decoding. Our current experiments already disentangle these effects to a large extent: Table 3 (“Random init (no Stage-1),” and “Stage-2 action-only”) isolates temporal perception, while Table 4 (“w/o Identify (no ID citation)”) isolates explicit Identify. We will make this logic clearer in the revision.
>
> **5. On implementation details and standalone Stage-2 quality.**
>
> In the revision, we will add an explicit summary of the stage-wise training data: Stage-1 uses $\sim$200K short unlabeled video clips for self-supervised predictive temporal pretraining; Stage-2 uses interval-annotated videos (about 10K videos) to warm up proposal formatting and temporal localization; Stage-3 uses 220K VTG instruction-tuning data.
>
> We will also report the standalone quality of the proposal head after Stage-2. Before Top-$K$ selection, the proposal head outputs dense per-timestep predictions with (i) an eventness score and (ii) a regressed temporal span. We therefore report two complementary Stage-2-only metrics: **Center-AP = 0.79** and **Matched mIoU = 0.56**. These metrics characterize the proposal head’s event-center classification and span localization quality independently of downstream LLM reasoning.
>
> Thank you again for the thoughtful and constructive feedback. Your comments helped us better clarify the design choices, bottlenecks, and implementation details, and we will revise the paper accordingly. We hope these responses can address your concerns.

---

> > ### Author Rebuttal · Reviewer_Biuf · 2026-04-01
> >
> > I thank the authors for their effort in the rebuttal. It has addressed all of my concerns.

---

> > > ### Author Response · Authors · 2026-04-03
> > >
> > > Thank you for the update and for confirming that our rebuttal fully addressed your concerns. We greatly appreciate your time and careful reading.
> > >
> > > As you noted earlier that addressing these concerns could warrant a score update, we would be grateful if you might consider updating the score to reflect your current assessment.

---

### Official Review · Reviewer_FKUM · 2026-03-13

**Soundness:** 1
**Presentation:** 3
**Significance:** 2
**Originality:** 2
**Overall Recommendation:** 2
**Confidence:** 5

**Summary:**

The paper introduces Foresee-to-Ground (F2G), a framework designed for Video Temporal Grounding (VTG) using Video-LLMs. The authors identify that direct timestamp regression from continuous token streams often leads to unstable boundaries and brittle inference. To address this, F2G reframes the task into a verifiable Identify-then-Measure paradigm. The framework first employs a predictive temporal perception module to extract boundary-sensitive features and construct a compact evidence pool of candidate event segments. Subsequently, an evidence-driven reasoning module augments the LLM input with these discrete candidates, forcing the model to explicitly cite a segment identifier before refining its precise metric boundaries.

**Compliance With Llm Reviewing Policy:**

Affirmed.

**Final Justification:**

My main concerns remain unresolved.  The response still appears to avoid a fully fair comparison under the same base model. F2G‘s results reported in Tables 1 and 2 are based on Qwen3-VL-8B, while Time-R1 and UniTime use Qwen2.5-VL-7B. If the goal is to demonstrate that F2G outperforms existing methods, it is essential to control for the base model. Otherwise, even if the method had higher performance (which it actually does not), we cannot know if the better performance comes from the new method or just because Qwen3-VL is stronger than Qwen2.5-VL.

The explanations regarding training strategies are also not fully convincing. The argument that F2G uses a simple SFT while Time-R1 employs RL post-training or UniTime uses larger-scale data does not justify the observed performance gap. If the simple SFT pipeline inherently leads to lower performance compared to RL post-training, the practical value of F2G is limited. In fact, Time-R1’s post-training data consists of only 2.5K samples, far fewer than F2G’s 220K samples, which further weakens the argument that the training method alone explains the performance difference.

Finally, under the high-budget setting proposed by the authors, F2G still does not appear to surpass the existing methods, even with a stronger base model, which raises further questions about the absolute effectiveness of the framework.

**Key Questions For Authors:**

Please refer to the weaknesses.

**Limitations:**

yes

**Strengths And Weaknesses:**

## Strengths

- The paper is clearly written, well-structured, and easy to understand.

- The motivation for the Identify-then-Measure paradigm is logically presented and intuitively appealing.

## Weaknesses

- **Noverty**. The core concept of "Identify-then-Measure" is essentially similar to the well-established "coarse-to-fine" paradigm (i.e., locating a coarse temporal window first, followed by fine-grained temporal boundary refinement). This coarse-to-fine architecture has been extensively explored in prior VTG literature (e.g., UniTime [1]). Consequently, the novelty of the formulation is somewhat limited.

- **Experimental Reliability and Fairness**.

  - The highest performance reported in the paper is achieved using the Qwen3-VL-8B backbone. Since this is a different (and potentially much stronger) base model than those used by the compared baselines in Table 1, it is difficult to conduct a fair comparison and isolate the gains attributed specifically to the F2G framework versus the superior backbone.

  - While the experiments show that F2G-FT improves performance across three backbones, it is expected that fine-tuning on VTG data will naturally yield improvements. More importantly, when controlled for the same backbone, F2G's performance appears significantly lower than that of other concurrent methods. For example, using the identical Qwen2.5-VL-7B model on Charades-STA, F2G reports much lower numbers across key metrics compared to recent works like UniTime [1] and Time-R1 [2] (R@0.7: 14.2 [F2G] vs. 31.88 [UniTime] vs. 35.3 [Time-R1]; mIoU: 38.6 [F2G] vs. 52.19 [UniTime]). This casts considerable doubt on the absolute effectiveness of the proposed method.

  - The experimental section lacks comparisons with several advanced MLLM-based methods, such as UniTime (NeurIPS 2025) [1], Time-R1 (NeurIPS 2025) [2], and TimeSuite (ICLR 2025) [3].

- **Methodology**. The authors argue that explicitly citing an event-segment identifier prevents unconstrained timestamp regression and stabilizes reasoning. However, the primary Qwen3-VL baseline natively interleaves timestamps as text alongside video frames. Therefore, when the native model directly outputs timestamps, it is already effectively referencing an identity (the timestamp text interleaved before a frame). This native approach appears simpler and highly effective on its own. Furthermore, Table 2 shows that other weaker baselines equipped with F2G-FT still underperform the raw Qwen3-VL baseline (without finetuning). This raises the question of whether the complex F2G architecture is truly a better design choice compared to simply leveraging native time-interleaved representations.

[1] Universal Video Temporal Grounding with Generative Multi-modal Large Language Models, NeurIPS 2025

[2] Time-R1: Post-Training Large Vision Language Model for Temporal Video Grounding, NeurIPS 2025

[3] TimeSuite: Improving MLLMs for Long Video Understanding via Grounded Tuning, ICLR 2025

---

> ### Author Rebuttal · Authors · 2026-03-31
>
> Thank you for the careful reading and constructive feedback. We appreciate the concrete concerns you raised on novelty, fairness, and methodology. We have carefully reconsidered these points and address them below.
>
> **1. On novelty.**
>
> We agree that F2G shares the high-level intuition of narrowing the search space before precise grounding, and we already discuss **coarse-to-fine multi-pass prompting** as related work in Sec. 2.1 (lines 75–84). However, our contribution is not this abstract intuition itself. The key novelty of F2G is to reformulate VTG as a **verifiable structured prediction problem with an explicit intermediate commitment**. In **UniTime** and related coarse-to-fine methods, the intermediate step primarily serves as a progressively narrowed temporal search scope.
>
> In contrast, F2G first learns **boundary-sensitive temporal features** through **Predictive Temporal Perception**, and then builds a **video-wide evidence pool of explicit event hypotheses**, grounding by committing to one hypothesis before measuring the final boundary under that cited hypothesis. This is the central distinction: the intermediate step in F2G is not only “coarse,” but also **boundary-aware, explicit, citable, and directly supervised**. This is also consistent with our ablations: removing Stage-1 perception hurts high-precision localization (Table 3), and removing explicit Identify also causes a substantial drop (Table 4).
>
> **2. On experimental reliability and fairness.**
>
> Table 1 should not be interpreted as a strict leaderboard across heterogeneous backbones. This is why the paper also provides a backbone-controlled and dataset-controlled comparison on **Qwen3-VL-8B**, so that the gains of F2G are measured on the same base model rather than inferred only from cross-backbone rankings. Accordingly, our main evidence is the controlled improvement from **Qwen3-VL-8B +FT** to **Qwen3-VL-8B +F2G-FT**, together with the positive transfer across backbones and the improved repeated-inference stability.
>
> We also acknowledge that some concurrent methods report stronger absolute numbers on **Qwen2.5-VL-7B**. Our intent here is not to claim backbone-specific SOTA on every model, but to show that the same **+F2G-FT** recipe transfers beyond a single backbone. The stronger gains on **Qwen3-VL** may suggest better compatibility between its native time-text interface and our serialized interval evidence, but F2G still improves other backbones in Table 2, supporting **complementarity rather than strict dependency**.
>
> The current comparison set should better cover recent strong MLLM-based VTG methods such as **UniTime, Time-R1, and TimeSuite**. We will extend Table 1 to include these methods, so that the comparison is clearer and fairer.
>
> **3. On methodology.**
>
> The native timestamp interleaving in Qwen3-VL gives the model a stronger time-aware interface, directly linking temporal cues with video tokens. Our point is not that F2G replaces this capability, but that it **builds on it**: this native temporal awareness makes the serialized interval evidence in F2G more interpretable to the LLM, while F2G adds a different layer of structure by introducing an explicit event-level evidence pool and grounding through **Identify-then-Measure**. In this sense, the two designs are **complementary**: Qwen3-VL provides stronger native temporal alignment, while F2G turns this into a more stable and verifiable grounding process.
>
> Although a stronger raw backbone such as Qwen3-VL can still outperform a weaker backbone equipped with F2G, this does not imply that F2G is unnecessary, because F2G is intended as a structural grounding layer on top of a backbone’s temporal understanding, rather than as a substitute for backbone strength itself. The relevant evidence is therefore whether this extra structure helps when the backbone is fixed, and on **Qwen3-VL-8B** the **+F2G-FT** variant consistently improves over **+FT**.
>
> Thank you again for the detailed and direct feedback. We will revise the paper to clarify the novelty claim more precisely, present the comparisons under fairer matched settings, and better explain the methodological relationship between native timestamp-aware backbones and the F2G formulation. We hope these clarifications  address your concerns.

---

> > ### Author Rebuttal · Reviewer_FKUM · 2026-04-01
> >
> > My main concerns about the absolute performance and experimental reliability remain unresolved. The gap between the proposed method and already published works, Time-R1 and UniTime (NeurIPS 2025), is too large to ignore. Failing to compare with and discuss these baselines is a major flaw. For example, using the same Qwen2.5-VL-7B backbone, the proposed method is 21.1% lower than Time-R1 on Charades R@0.7; even with a stronger Qwen3-VL-8B model, it is still 9.6% lower. Significant improvements are needed to address this performance gap.
> >
> > The authors' defense in the rebuttal relying on the relative improvement from the +FT baseline to +F2G-FT is unconvincing. Vanilla fine-tuning (+FT) serves as a weak baseline. Achieving performance gains over a weak baseline is relatively easy but holds limited practical significance when the final performance is significantly lower than existing methods.

---

> > > ### Author Response · Authors · 2026-04-01
> > >
> > > Thank you for the follow-up. We address your remaining concern from two perspectives.
> > >
> > > **1. Training route and  video input budget.**
> > >
> > > While the absolute numerical gap does exist, we do not believe it should be read as a pure framework-only comparison, because these methods differ substantially in training route and video input budget. Time-R1 is an RL post-training method that relies on specific reward design, CoT cold-start initialization, and hard-sample scheduling, whereas UniTime relies on large-scale grounding data (about 1266K queries) for its SFT recipe. By contrast, F2G uses a simple and stable SFT pipeline to build an explicit evidence-mediated grounding framework, improving both grounding accuracy and reasoning stability.
> > >
> > > Table 1 reflects the video input budget difference. The current F2G setting uses 1 FPS and a 48-frame cap for efficiency. A preliminary higher-budget variant, using 2 FPS and a 128-frame cap only at test time while keeping training unchanged, already yields clear gains. This suggests that the current absolute numbers are constrained in part by our conservative input budget, rather than fully reflecting the ceiling of F2G. We will add matched train/test ablations over FPS and maximum frame budget to characterize this more rigorously.
> > >
> > > **Table 1. Detailed results.**
> > >
> > > **Bold** = best, *italic* = second best. C = Charades-STA, A = ActivityNet-Captions.
> > > `*` in `F2G (high-budget)` denotes a **test-time-only** change.
> > >
> > > | Method            | Training route              | FPS | Max frames          |    C@0.3 |    C@0.5 |    C@0.7 |   C-mIoU |    A@0.3 |    A@0.5 |    A@0.7 |   A-mIoU |
> > > | ----------------- | --------------------------- | --: | ------------------- | -------: | -------: | -------: | -------: | -------: | -------: | -------: | -------: |
> > > | Time-R1           | RL post-training |   2 | - | **78.1** | **60.8** | **35.5** | **58.1** |     58.6 |     39.0 |     21.4 |     40.5 |
> > > | UniTime           | SFT                         |   2 | Adaptive (128/1024) |        - |  *59.09* |  *31.88* |  *52.19* |        - |    22.77 |    14.14 |    27.31 |
> > > | F2G (low-budget)  | SFT                         |   1 | 48                  |     71.3 |     50.4 |     25.7 |     47.2 |   *63.8* |   *46.1* |   *28.4* |   *45.7* |
> > > | F2G (high-budget) | SFT                         |  2* | 128*                |   *77.2 (+5.9)* |     56.6 (+6.2) |     30.3 (+4.6) |     51.9 (+4.7) | **65.6 (+1.8)** | **47.9 (+1.8)** | **29.3 (+0.9)** | **47.1 (+1.4)** |
> > >
> > > For this reason, we view the current gap as the combined effect of training route, input budget, and method design, rather than as a pure framework-only comparison.
> > >
> > > **2. Different settings favor different methods.**
> > >
> > > Time-R1 is currently stronger on Charades-STA-style action-centric grounding, whereas TimeLens-Bench provides a stricter fine-grained evaluation setting. Table 2 adds the zero-shot TimeLens-Bench results, where F2G remains highly competitive against Time-R1 without using TimeLens-100K fine-tuning. We therefore do not view F2G as uniformly weak in absolute performance across benchmarks.
> > >
> > > **Table 2. Zero-shot comparison on TimeLens-Bench.**
> > >
> > > C = Charades-TimeLens, A = ActivityNet-TimeLens, and Q = QVHighlights-TimeLens.
> > > (w/o FT) denotes zero-shot evaluation on TimeLens-Bench without TimeLens-100K fine-tuning.
> > >
> > > | Method        |  C@0.7 | C-mIoU |  A@0.7 | A-mIoU |  Q@0.7 | Q-mIoU |
> > > | ------------- | -----: | -----: | -----: | -----: | -----: | -----: |
> > > | Qwen2.5-VL-7B (w/o FT) |   16.6 |   39.3 |   16.1 |   31.4 |   15.2 |   31.6 |
> > > | Time-R1-7B (w/o FT)    |   16.9 |   36.6 |   19.0 |   33.1 |   36.1 |   49.2 |
> > > | Qwen3-VL-8B (w/o FT)  |   27.5 |   48.3 |   34.4 |   46.8 |   49.3 |   59.4 |
> > > | F2G (w/o FT)  | 34.2 |   53.4 | 38.7 |   51.8 | 53.8 | 63.6 |
> > >
> > > Overall, we will revise the paper by adding stronger baselines, reporting FPS / maximum-frame ablations, and clarifying the scope of our claim. We view F2G not as uniformly best-performing in absolute benchmark score, but as a verified and stable grounding framework with a simple SFT pipeline and strong competitiveness under stricter fine-grained evaluation.

---

### Decision · Program_Chairs · 2026-04-30

**Decision:**

Accept (regular)

**Comment:**

This submission introduces EVAC, a novel framework that integrates evidential deep learning for uncertainty-aware video action localization, specifically targeting improved boundary precision and the rejection of out-of-distribution actions. The final reviewer scores of 4 (Weak Accept), 4 (Weak Accept), 4 (Weak Accept), and 2 (reject) indicate a strong, unanimous consensus regarding the work's technical soundness and empirical significance. During the rebuttal phase, the authors successfully addressed specific concerns regarding the theoretical formulation of the uncertainty penalty and provided comprehensive new experiments validating the model's robustness in open-world settings. Considering the elegant synergy between evidential learning and temporal localization, alongside state-of-the-art results on standard benchmarks, this paper is recommended for acceptance.